# Mechanism of APTX nicked DNA sensing and pleiotropic inactivation in neurodegenerative disease

Percy Tumbale[1,†,§], Matthew J Schellenberg[1,†,§], Geoffrey A Mueller[1,†,§], Emma Fairweather[2], Mandy Watson[2], Jessica N Little[1,§], Juno Krahn[1,§], Ian Waddell[2], Robert E London[1,§] & R Scott Williams[1,*,§]

## Abstract

The failure of DNA ligases to complete their catalytic reactions generates cytotoxic adenylated DNA strand breaks. The APTX RNA-DNA deadenylase protects genome integrity and corrects abortive DNA ligation arising during ribonucleotide excision repair and base excision DNA repair, and *APTX* human mutations cause the neurodegenerative disorder ataxia with oculomotor ataxia 1 (AOA1). How APTX senses cognate DNA nicks and is inactivated in AOA1 remains incompletely defined. Here, we report X-ray structures of APTX engaging nicked RNA-DNA substrates that provide direct evidence for a wedge-pivot-cut strategy for 5′-AMP resolution shared with the alternate 5′-AMP processing enzymes POLβ and FEN1. Our results uncover a DNA-induced fit mechanism regulating APTX active site loop conformations and assembly of a catalytically competent active center. Further, based on comprehensive biochemical, X-ray and solution NMR results, we define a complex hierarchy for the differential impacts of the AOA1 mutational spectrum on APTX structure and activity. Sixteen AOA1 variants impact APTX protein stability, one mutation directly alters deadenylation reaction chemistry, and a dominant AOA1 variant unexpectedly allosterically modulates APTX active site conformations.

**Keywords** APTX; Ataxia Oculomotor Apraxia 1; DNA repair; missense mutation; X-ray crystallography
**Subject Categories** DNA Replication, Repair & Recombination; Molecular Biology of Disease; Structural Biology
**The EMBO Journal (2018) 37: e98875**

## Introduction

DNA ligation is a central process in biology that finalizes genome maintenance metabolic processes including DNA replication, recombination, and DNA repair. Eukaryotic DNA ligases catalyze ligation via a three-step, ATP-dependent reaction. First, the DNA ligase active site lysine is adenylated. Second, the adenylate is transferred to a DNA 5′ phosphate to facilitate the third nick-sealing step. Third, nucleophilic attack of a 3′-OH on the activated 5′-adenylate facilitates phosphodiester bond formation, and sealing of the DNA break (Pascal *et al*, 2004; Tomkinson *et al*, 2006). Environmental and metabolic sources of DNA damage prompt "abortive ligation", the failure of ligase to complete step 3, and generation of 5′-adenylated (5′-AMP) DNA strand breaks (Ahel *et al*, 2006; Andres *et al*, 2015; Schellenberg *et al*, 2015) (Fig 1A). Triggers of abortive ligation include RNA-DNA junction intermediates in ribonucleotide excision repair (RER) (Tumbale *et al*, 2014), single-strand breaks with oxidative DNA base damage (e.g. 3′-8-oxo-dG) (Parsons *et al*, 2005; Ahel *et al*, 2006; Harris *et al*, 2009), and DNA nicks bearing 5′ deoxyribose phosphate groups, such as those generated by base excision repair (BER) (Rass *et al*, 2007). In these contexts, it is hypothesized that DNA ligase exacerbates genome instability with the creation of complex 5′-adenylated damage comprised of the instigating lesion, compounded by adenylation of the DNA 5′-terminus.

The apratoxin (APTX) RNA/DNA polynucleotide deadenylase directly reverses 5′-AMP damage (Fig 1A) (Ahel *et al*, 2006; Harris *et al*, 2009; Reynolds *et al*, 2009; Tumbale *et al*, 2011, 2014) and associates with DNA repair scaffolds XRCC1 (Cherry *et al*, 2015) and XRCC4 (Breslin & Caldecott, 2009). Mutational inactivation of APTX is associated with elevated levels of oxidative DNA damage (Hirano *et al*, 2007; Harris *et al*, 2009), and increased DNA damage following treatment with the anticancer topoisomerase inhibitor CPT (Mosesso *et al*, 2005), and *S. pombe* Aptx mutants are sensitive to 4NQO (Deshpande *et al*, 2009). Genetic evidence implicates the budding yeast APTX homolog Hnt3p in repair of RNA-triggered

1 Genome Integrity and Structural Biology Laboratory, Department of Health and Human Services, National Institutes of Environmental Health Sciences, US National Institutes of Health, Research Triangle Park, NC, USA
2 Drug Discovery Group Cancer Research UK Manchester Institute, Manchester, UK
   *Corresponding author. Tel: +1 984 287 3542; E-mail: williamsrs@niehs.nih.gov
   †These authors contributed equally to this work
   §This article has been contributed to by US Government employees and their work is in the public domain in the USA

abortive ligation during ribonucleotide excision repair (RER) (Tumbale *et al*, 2014), as well as alkylation and oxidative DNA damage repair pathways (Daley *et al*, 2010). Aptx-deficient mice expressing mutant superoxide dismutase (SOD1) show cellular survival defects in cultured cells (Carroll *et al*, 2015), and the genetic combination of *Aptx* and *Tdp1* (tyrosyl-DNA phosphodiesterase 1) knockout results in global defects in repair of oxidative and alkylation-induced DNA damage (El-Khamisy *et al*, 2009). The consequences of APTX dysfunction in humans are severe. *APTX* mutations are linked to the progressive neurodegenerative diseases ataxia with Oculomotor Apraxia 1 (AOA1) (Date *et al*, 2001; Moreira *et al*, 2001), ataxia with coenzyme Q10 (coQ10) deficiency (Quinzii *et al*, 2005), and a multiple system atrophy resembling Parkinson's disease (Baba *et al*, 2007). AOA1 mutations map to the APTX catalytic domain (Barbot *et al*, 2001; Moreira *et al*, 2001; Shimazaki *et al*, 2002; Le Ber *et al*, 2003; Tranchant *et al*, 2003; Criscuolo *et al*, 2004, 2005; Ito *et al*, 2005; Mosesso *et al*, 2005; Baba *et al*, 2007; Yokoseki *et al*, 2011) and cause variable age of disease onset. Yet, how these mutations impact the APTX structure and its polynucleotide deadenylase functions remains largely unknown.

Previous molecular structural interrogations of *S. pombe* (Tumbale *et al*, 2011; Chauleau *et al*, 2015) and human (Tumbale *et al*, 2014) APTX homologs by us and others have resolved how APTX engages blunt DNA duplex structures, and provided a basis for understanding the APTX direct-reversal DNA deadenylation reaction. However, the presumed cognate substrates for APTX that arise during ribonucleotide excision repair (RER) and base excision repair are nicked DNA and RNA-DNA junctions. How APTX senses DNA nicks and processes adenylation damage in these contexts remains undefined. It is also unknown whether the APTX structure and activity is regulated, and if so, how disease states might impact such regulation. To help resolve these questions, we have secured molecular snapshots of APTX in complex with nicked RNA-DNA substrates and investigated its conformations throughout its reaction cycle using nuclear magnetic resonance (NMR) spectroscopy. We uncovered a DNA nick-induced fit active site assembly mechanism that modulates APTX active site loop conformations. Furthermore, results from a comprehensive structural and functional characterization of AOA1 mutants provide a framework for understanding APTX inactivation in neurodegenerative disease. We find AOA1 mutations impair APTX function by impacting protein folding, altering active site chemistry, or by perturbing ligand-dependent APTX conformational changes.

# Results

### Molecular architecture of APTX bound to nicked RNA-DNA

To visualize APTX bound to a cognate reaction product harboring nicked RNA-DNA and a cleaved AMP lesion, we surveyed 80 combinations of protein/RNA-DNA complexes for co-crystallization. Our combinatorial crystallization approach involved varying upstream and downstream duplex lengths surrounding a DNA nick, varying position of the nick, as well as concatemerization of DNA nicks within a single target nicked RNA/DNA substrate (Fig 1B). We obtained crystals (Appendix Fig S1A) diffracting to 2.4 Å resolution that contain two APTX catalytic domain (APTX[cat], residues

165–342) protomers engaging a doubly nicked target structure (Fig 1B and C). Successful crystallization strategies ultimately utilized a concatemerized nick substrate with a 6 bp upstream and 8 bp downstream duplexes (Fig 1B). In addition, a 5′-ribonucleotide is positioned at the nick junction. In this configuration, the structural snapshots captured in our crystal structures are representative of APTX bound to a nicked RNA-5′-3′-DNA junction reaction product complex formed by APTX during repair of abortive ligation products created during ribonucleotide excision repair (RER) (Tumbale *et al*, 2014; Schellenberg *et al*, 2015).

The structure reveals how APTX induces large-scale DNA duplex substrate distortions in order to access the 5′-terminal adenylated lesion in the context of a nicked duplex. In each of the complexes captured in our crystals, abrupt DNA bending is governed by the APTX amino-terminal helix, as previously hypothesized (Tumbale *et al*, 2011). This structure illuminates how APTX senses DNA damage through two discontinuous nucleic acid binding sub-sites, dictated by both the HIT and Znf domains which together collaborate to sense the nick. The electron density for the bound nick target is well defined in the crystals (Fig 1C). Although the doubly nicked RNA-DNA sequence is symmetrical, about an 8 bp double-stranded palindromic region, in three-dimensional space this twofold symmetry, is broken. Two nick-bound APTX molecules bind the upstream region of the nick with distinct conformations (Appendix Fig S1B, and Materials and Methods). In both binding modes captured, APTX-directed DNA wedging and penetration into the base stack results in the redirection of the base stack approximately orthogonal to the helical axis of the bound downstream region.

The major molecular interface at the DNA nick is mediated by the amino-terminal HIT domain α-helix (α1) that infiltrates the DNA base stack. Here, α1 serves as a doubly barbed "wedge" that splays the base stack apart (Figs 1D and 2A, orange helix). The planar rings of His166 and Trp167 together redirect the DNA duplex, imparting a ~90° bend to the substrate RNA-DNA. The protein displaces the upstream half of the nick and unwinds both the 3′ and 5′ sides of the nick. This has two effects: First, extraction of the 5′-terminus facilitates positioning of the adenylated lesion into the active site; and second, disruption of the 3′-terminal side of the nick exposes the 3′ end.

### DNA-induced fit APTX conformational changes

Previous X-ray structures of APTX bound to blunt-ended DNA displayed two major APTX conformations differing by a conformational change of α1, a series of linked rearrangements of the histidine triad (HIT, HxHxH) substrate engagement loop, and additional compensating movements throughout the fused HIT-Znf catalytic core (Tumbale *et al*, 2014). In a "closed", and catalytically competent conformer, α1 packs against the HIT-loop and aligns the active site nucleophile His260 for attack on the 5′-AMP phosphorous atom. In comparison, a catalytically incompetent "open" conformer is typified by partial disengagement of the α1 helix from the HIT-loop, and a misaligned active site (Appendix Fig S2A). Structural superpositions indicate the present nick-bound APTX protomers are both found in the closed conformation, with the active site poised for catalysis (Appendix Fig S2B). Furthermore, in the DNA nick-bound forms, His166 (α1) stacks against the upstream duplex (Fig 2A).

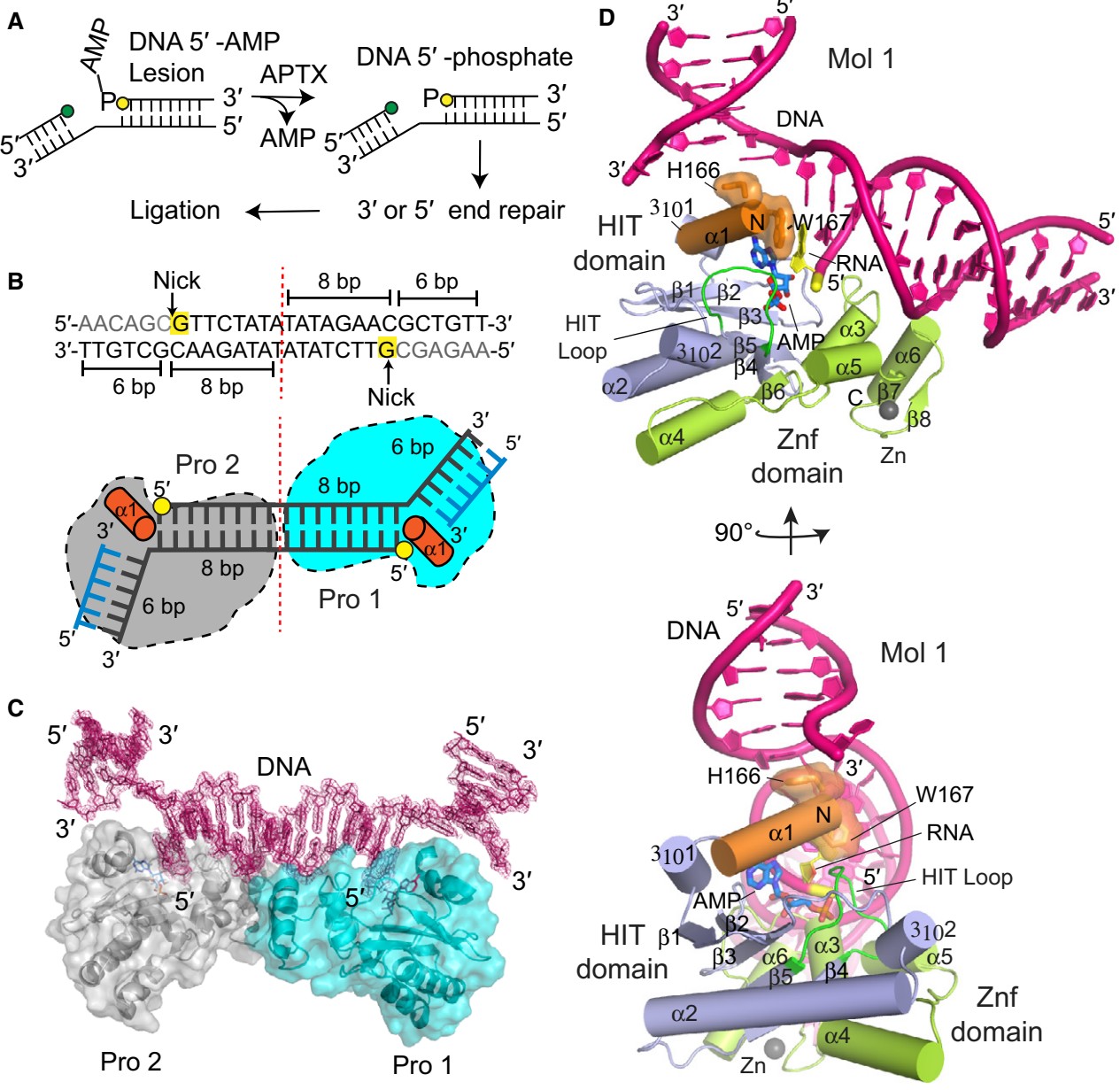

**Figure 1. X-ray structure of hAPTX-nicked-RNA-DNA complex.**

A  Schematic presentation of APTX 2-step deadenylation mechanism.

B  Concatemerized nicked RNA-DNA duplex substrate bound in the structure of hAPTX-nicked-RNA-DNA complex. The 6 bp upstream region is shown in blue, 8 bp downstream region, and the complementary template strand in black, 5′ ribonucleotide positioned at the nick junction in yellow, and N-terminal α1-helix in orange.

C  Transparent surface representation of two APTX protomers (Pro1, cyan and Pro2, gray) bound to a single RNA-DNA molecule containing two nick sub-sites in each asymmetric unit. Simulated annealing omit 2Fo-Fc electron density for the bound nicked RNA-DNA (magenta) illustrates APTX engagement to DNA nick induces large-scale DNA distortions at the nick.

D  X-ray structure of hAPTX-nicked-RNA-DNA complex. N-terminal α1-helix is colored orange, HIT domain in light blue, Znf domain in light green, DNA in pink, RNA in yellow, AMP in teal, and HIT-loop in dark green.

Modeling the opened state with this DNA shows it would clash in the open conformation, suggesting the interaction of His166 with the upstream duplex reinforces a closed and catalytically competent active site. Collectively, these observations suggest that DNA substrate engagement regulates conformations of the HIT wedge helix and, in turn, active site assembly.

Difference distance matrix plot (DDMP) analysis (Appendix Fig S3) and a structural interpolation comparing the open and closed APTX conformational states (Movie EV1) reveals significant overall rearrangements of the HIT-Znf scaffold. We thus hypothesized that substrate binding and catalysis are intimately associated with α1 conformational change. To test this, we probed the structural

response of full-length human APTX (hAPTX[FL]) to substrate and AMP lesion binding using limited proteolysis coupled to mass spectrometry. In the absence of DNA and the AMP lesion, chymotrypsin protease efficiently degrades hAPTX[FL], yielding a metastable fragment (fragment "C1" aa 152–342, blue, Fig 2B, and Appendix Fig S4). C1 is digested further by cleavages at Trp167 (C2) and Leu171 (C3) in the N-terminal helix α1, revealing that in the absence of DNA, relative to the HIT-Znf core, helix α1 is flexible, accessible to protease, and therefore partially unfolded in solution. Addition of blunt-ended DNA or a nicked DNA oligonucleotide in the presence of AMP resulted in proteolytic protection of α1 (sites C2 and C3), consistent with a DNA-induced conformational ordering of α1 (Fig 2B). Similar proteolytic protection was observed with incubation with nicked DNA, adenosine (Ade), and orthovanadate ($VO_4^{3-}$) under solution conditions that have been previously demonstrated to covalently trap an APTX transition-like state (Tumbale *et al*, 2014; Fig 2B, lanes 11–15). That APTX was not protected from proteolysis when incubated with relaxed circular plasmid DNA indicates that helix α1 is specifically responsive to DNA damage binding at DNA ends and nicks (Fig 2B, lane 21–25 Appendix Fig S4). Moreover, an AOA1 active site mutant K197Q which impacts the active site DNA and AMP binding pocket displays impaired DNA-dependent proteolytic protection (Fig 2B, lanes 27–45). Thus, DNA-binding linked α1 conformational change is altered by AOA1 mutations linked to neurological disease.

## Global conformational responses of the APTX catalytic domain during catalysis

To better define the response of the APTX catalytic domain (APTX[cat]) to substrate binding in solution, we investigated the behavior of $^{13}CH_3$-Met-labeled APTX[cat] by NMR. The $^{13}CH_3$-Met labels provide seven probes distributed throughout APTX[cat] located proximal to α1 (Met164, Met175, and Met180), the HIT-loop (Met256), and at three added positions remote from α1 and the active center (Met227, Met309, and Met296) (Fig 2C and Appendix Table S1). These probes are also coincident with regions shown to undergo motion in X-ray structures (Appendix Fig S3). Distinct $^{13}CH_3$-Met resonance shifts are observed for apo (unliganded structure) compared with blunt-ended or nicked DNA substrates (DNA in twofold molar excess; Fig 2D).

The $^{13}CH_3$ shifts for Met164 are close to those expected for a methionine residue in a random coil (Butterfoss *et al*, 2010),

consistent with the origin of this residue as a residual fragment from an N-terminal His-tag. Nevertheless, the M164 $^1H$ shift is sensitive to the presence of nicked DNA, most probably as a result of the stacking of nearby Trp167 against the rG1 residue at the nick. As a result of its position near the N-terminus of helix α1, this observation underscores a key role for α1 in DNA nick recognition. Met175/Met180 (α1) and Met256 (HIT-loop) also display substrate-specific chemical shift perturbations. The large, downfield $^1H$ shift of ~0.4 ppm for M256 is consistent with its position at the edge of the Trp167 indole side chain in the nicked DNA complex (Appendix Table S1, Fig 2D). Consistent with these substrate-dependent conformational responses, kinetic analysis of APTX deadenylation reaction (Fig 2F) yielded distinct parameters for APTX deadenylase activity on blunt ($K_m = 37.1$, $k_{cat} = 0.51/s$) versus nicked ($K_m = 17.1$ nM, $k_{cat} = 0.38/s$) substrates, with APTX displaying superior catalytic efficiency on nicked DNA ($k_{cat}/K_m = 0.022$ nicked, $k_{cat}/K_m = 0.014$ blunt DNA).

To obtain further insight into catalysis-related conformational changes, we measured the $^{13}CH_3$-Met resonance shifts to compare (i) Apo (Black, Fig 2E), (ii) reaction transition state (blue, Fig 2E), and (iii) product states (purple, Fig 2E). Comparison of $^{13}CH_3$-Met spectra shows that major changes in the molecular environment of Met256 (HIT-loop), Met175, and Met180 (α1) occur during the APTX reaction cycle. Strikingly, the methyl resonances of Met227, Met309, and to a lesser extent, Met296, corresponding to residues located removed from the active site (> 10–15 Å, Appendix Table S1) and substrate binding regions of the enzyme, also exhibit significant substrate (Fig 2D) and reaction-state (Fig 2E)-dependent chemical shift behaviors. The chemical shift sensitivity of residues remote from the active site may be amplified by proximity to shift-inducing aromatic residues. For instance, the M227 shift may be responsive to small positional variations relative to Y250. Overall, these NMR solution results are consistent with our X-ray structures and proteolysis indicating conformational change in α1 and the HIT-loop is coincident with a global APTX conformational response to DNA engagement. These observations further implicate a DNA regulated active site assembly/disassembly cycle in catalysis.

## APTX mutations variably impact protein folding and activity

The crystallographically defined APTX intermediate states and NMR characterized ligand-dependent conformational changes provide a framework for understanding the molecular consequences of the

---

**Figure 2.   DNA-induced conformational ordering of N-terminal α1-helix.**

A   Molecular details of APTX-DNA nick interface illustrate the N-terminal α1-helix wedges into the DNA base stack. His166 and Trp167 stack against the bases at the nick, bending the DNA duplex at a ~90° and unwinding both the 3′ and 5′ ends of the damaged strand.

B   Limited chymotryptic proteolysis of APTX. The unliganded full-length APTX was proteolysed to C1 fragment (blue) mapped to the N-terminus of the HIT domain. The C1 fragment was further degraded to C2 (magenta) and C3 (green), and mapped to the N-term α1-helix (orange box). Addition of blunt-ended DNA, nicked DNA, and transition mimic DNA (adenosine (Ade) and orthovanadate ($VO_4^{3-}$)) substrates resulted in proteolytic protection of C1. APTX was not protected from proteolysis when incubated with relaxed circular plasmid DNA. AOA1 K197Q active site mutant that is severely defective in DNA binding displays impaired DNA-end-dependent proteolytic protection.

C   Locations of [methyl-$^{13}C$]-labeled methionine residues (cyan) mapped in the catalytic core of APTX.

D   Overlays of assigned $^1H$-$^{13}C$ HSQC spectra of [methyl-$^{13}C$]methionine-labeled wild-type APTX in unliganded state (black), blunt-ended RNA-DNA substrate bound state (cyan), and nicked RNA-DNA substrate bound state (red). The arrows indicate resonance shifts in response to DNA substrates.

E   Overlays of assigned $^1H$-$^{13}C$ HSQC spectra of [methyl-$^{13}C$]methionine-labeled wild-type APTX in unliganded state (black), adenosine-$VO^3$-RNA-DNA transition mimic state (blue), and nicked AMP and RNA-DNA product bound state (purple). The arrows indicate resonance shifts in response to RNA-DNA substrates.

F   Kinetic parameters of APTX processing of 5′-adenylated RNA-DNA substrates, nicked (red) and blunt-ended (cyan). Mean ± 1 SD (*n* = 3 technical replicates) is displayed.

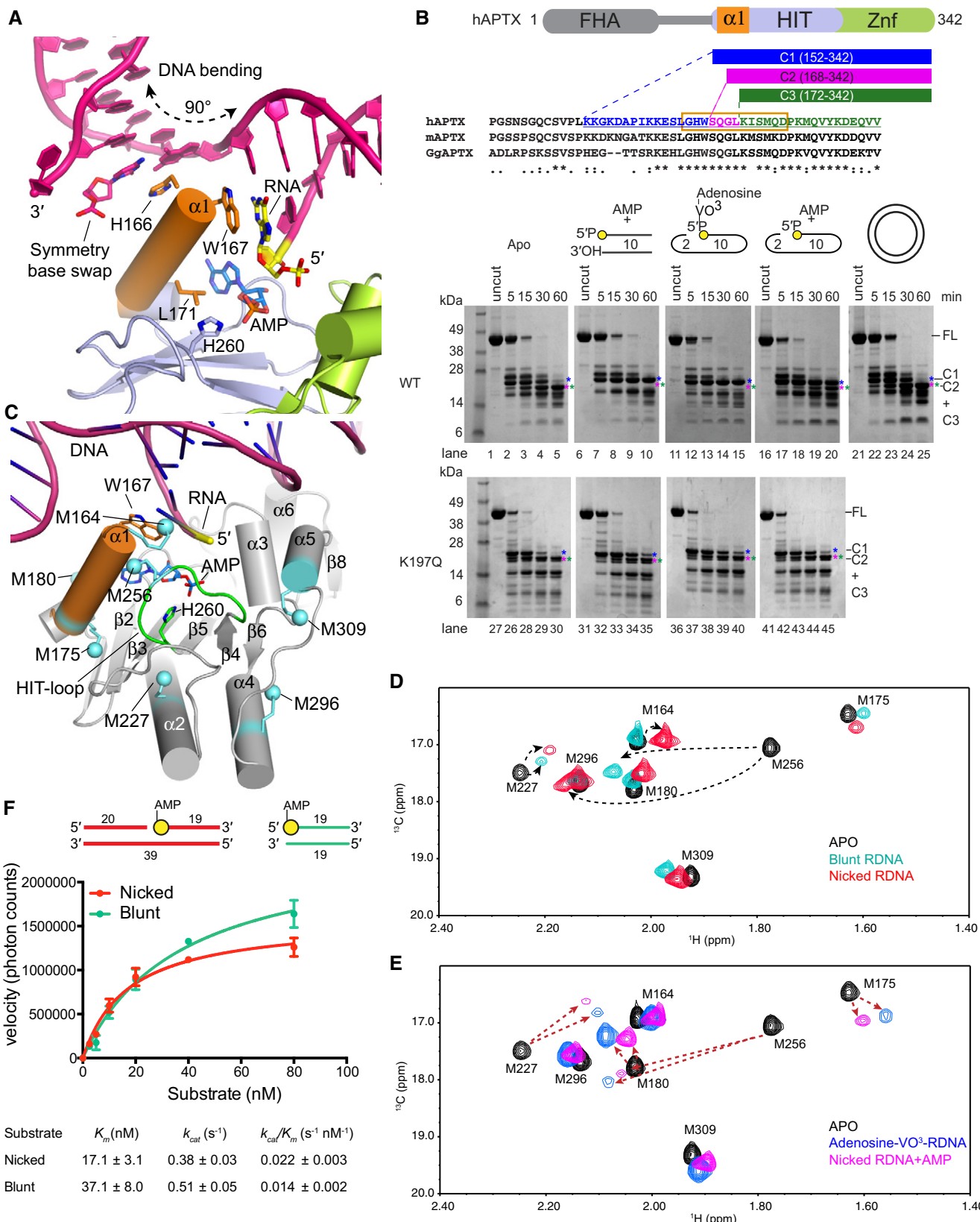

**Figure 2.**

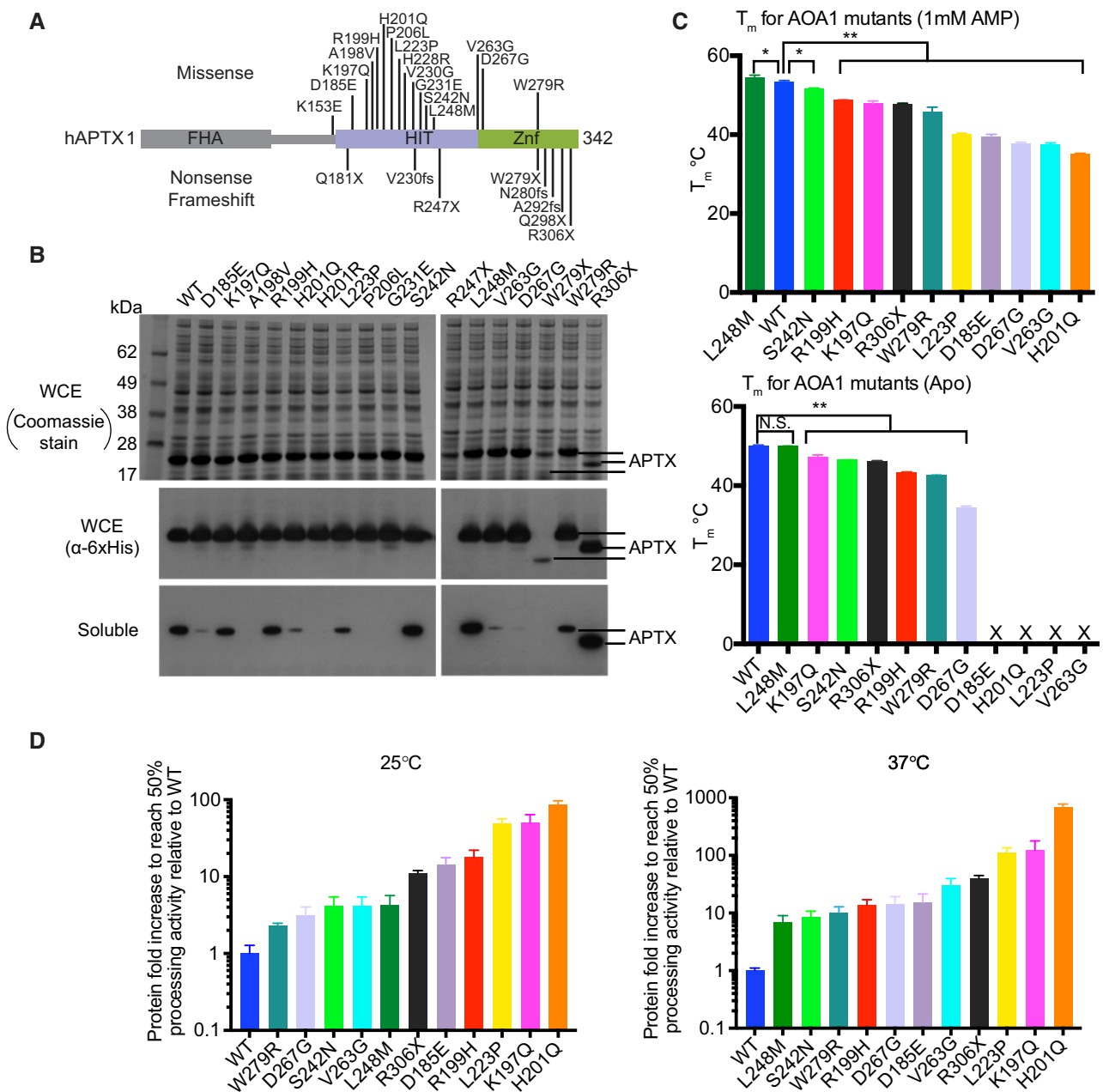

**Figure 3. AOA1 mutagenic effects on APTX solubility, stability, and catalytic activity.**

A  APTX mutations in AOA1 map to the catalytic core of APTX comprising the HIT (blue) and Znf (green) domains.
B  Solubility of 17 AOA1-linked APTX missense and nonsense mutants expressed in *E. coli* was assessed by Western blotting. With the exception of two nonsense variants (R247X and W279X), all proteins expressed at comparable levels. The soluble protein fractions revealed marked variability in solubility.
C  Thermal melting profile of APTX mutants. Unfolding of APTX proteins was monitored using SYPRO Orange fluorescent dye. The midpoint temperature ($T_m$) for APTX wild-type and mutants in unliganded (bottom) and AMP bound (top) conditions is displayed. X = unable to determine $T_m$, *P-value < 0.05, **P-value < 0.01, N.S. = not-significant, two-tailed *t*-test. Mean ± 1 SD (three technical replicates) is shown.
D  Deadenylation activity of APTX mutants. Tenfold dilutions of APTX mutant proteins were tested for deadenylation on a TAMRA labeled 5'-adenylated nicked RNA-DNA substrate at 25 and 37°C. Reaction products were resolved on 15% TBE-urea denaturing gels and visualized by fluorescence scan. Fold increase in protein to reach 50% activity relative to wild-type APTX is displayed. Mean ± 1 SD (3 technical replicates) is displayed.

many known AOA1-linked missense mutations. A subset of APTX mutants have been demonstrated to harbor defects in a weak APTX AMP-lysine hydrolase activity (Seidle *et al*, 2005). However, for the majority of AOA1 variants, the molecular and functional impacts on

APTX RNA-DNA deadenylase activity have not been assessed. Several AOA1 mutations are single amino acid substitutions within the protein core or lead to premature protein truncations (Fig 3A and Appendix Fig S5). We thus evaluated the impact of AOA1

   

mutations on protein folding. First, we tested the solubility of 17 AOA1-linked missense and nonsense mutants expressed in *E. coli*. With the exception of two nonsense variants (R247X and W279X), all proteins expressed at comparable levels as assessed by Western blotting to a 6x-His-tag in the recombinant protein (Fig 3B). In contrast, analysis of the soluble protein fractions revealed marked variability in solubility. Six mutants (A198V, H201R, P206L, G231E, R247X, W279X) were entirely insoluble, revealing these substitutions are poorly tolerated in the APTX structure. Examination of the molecular environments of these amino acid positions (Appendix Fig S5) shows these mutations are likely to disrupt folding of the HIT domain. The remaining 11 AOA1 variants (D185E, K197Q, R199H, H201Q, L223P, S242N, L248M, V263G, D267G, W279R, R306X) (Fig 3B) display wide ranging solubility, suggesting these mutations differentially impact protein stability and activity.

To further define the molecular basis for AOA1 defects, we purified soluble variants (Fig 3B, Appendix Fig S6) and measured protein thermal stabilities and RNA-DNA deadenylase activities (Fig 3C, Appendix Figs S6 and S7A, and Dataset EV1). Thermal shift assays derive a thermal melting point transition from fluorescence of Sypro Orange dye binding to the protein hydrophobic core as the protein undergoes heat-induced unfolding, in the presence or absence of bound ligands (Huynh & Partch, 2015) (Fig 3C). With the exception of L248M, all mutants displayed statistically significant decreases in protein thermal stability compared to WT. In the absence of ligand, accurate $T_m$ values for D185E, H201Q, V263G, and L223P could not be determined due to their intrinsic instability under the conditions examined. Addition of AMP (the APTX deadenylation reaction product) induced a positive thermal stability shift for WT ($T_m$-Apo = 50.1°C, $T_m$-AMP = 53.6°C). All AOA1 mutants, including those whose $T_m$ could not be determined without ligand, were also stabilized by AMP. Interestingly, L248M is more thermostable than WT in the presence of AMP.

To test the hypothesis that AOA1 mutations compromise APTX RNA-DNA processing capacity, we evaluated 5′-adenylated RNA-DNA reversal activity of purified mutant proteins. Two temperatures (22 and 37°C) were used to test temperature-dependent effects on catalysis (Fig 3D, Appendix Figs S6A and C, and S7B). This analysis shows that to varying degrees, all mutants were impaired in their ability to process 5′-adenylated RNA-DNA substrates. Consistent with stability defects, a majority of the AOA1 mutants (K197Q, S242N, R306X, W279R, D267G, H201Q, V263G, L223P) displayed a ~2- to 8-fold more severe catalytic impairment at 37°C compared to 22°C (Fig 3D and Appendix Fig S6C).

## Molecular bases for APTX inactivation in AOA1

Given the complex differences in the effects on thermal stability and catalytic activity, and the variable age of onset conferred by AOA1 mutations, we sought to better define the root molecular causes of AOA1-linked APTX defects. We utilized a blunt-ended substrate for co-crystallization with mutant APTX (Tumbale *et al*, 2014). This substrate yields crystals that grow rapidly (within 24 h) and typically diffract to higher resolution compared to nicked-DNAs whose protracted crystallization times proved intractable for structural analysis of destabilizing mutations. For five amenable variants (V263G, H201Q, S242N, R199H, and L248M), we successfully crystallized and determined X-ray structures of mutant proteins as ternary complexes with RNA-DNA and AMP (Fig 4, Appendix Table S2). The remaining mutants precipitated rapidly and could not be successfully crystallized under the conditions examined.

*V263G*: Val263 maps to the β-strand bearing the APTX catalytic nucleophile His260 and is a hydrophobic core residue engaging in Van der Waals interactions with neighboring Leu261 (β5), Val204 (β3), Ile232 (α2), His228 (α2), Phe246 (β4), and Val190 (β2). The structure of V263G (at 2.90 Å, Fig 4A–C) reveals the mutation creates a sizeable cavity in the hydrophobic core corresponding to loss of the valine side chain. Similar to core cavitation mutants of BRCA1 (Williams *et al*, 2003, 2004), V263G has severe impacts on stability (Fig 3C, and Appendix Fig S6C) and temperature-dependent activity, being ~7-fold more impaired for deadenylation activity at 37°C compared to 22°C (Fig 3D, Appendix Figs S6A and C).

*H201Q*: While His201 falls outside of the so-called HIT domain histidine triad motif, it is highly conserved in HIT domain

**Figure 4. X-ray structures of AOA1 mutants.**

A Omit 2Fo-Fc electron density for the V263G mutant.

B X-ray structure of the V263G mutant (blue) overlaid upon wild-type APTX (orange). Val263 maps to β5 and is a hydrophobic core residue engaging in close Van der Waals interactions with neighboring hydrophobic residues. The structure of V263G shows the mutation creates a large cavity in the hydrophobic core, resulted from the loss of the valine side chain.

C Mesh space filled presentation illustrates a cavity in APTX hydrophobic core created by the V263G mutation.

D Omit 2Fo-Fc electron density for the H201Q mutant.

E X-ray structure of the H201Q mutant (blue) overlaid upon wild-type APTX (orange). His201 participates in a charge relay mechanism that stabilizes the transition state of the APTX deadenylation reaction.

F Schematic presentation of the APTX active site. A charge relay consists of the main-chain amide of Lys197, His201, and His262 that hydrogen bonds to the scissile phosphate of the 5′-adenylate phosphoanhydride. The structure of H201Q reveals the disruption to the charge relay network.

G Omit 2Fo-Fc electron density for the S242N mutant.

H X-ray structure of the S242N mutant (blue) overlaid upon wild-type APTX (orange). Ser242 maps to a surface loop. The native loop conformation in wild-type APTX is stabilized through hydrogen bond between Ser242 hydroxyl and Leu244 main-chain amide. The serine to asparagine mutation causes rearrangements of side-chain contacts resulting in an altered loop structure with a 1.3 Å shift at the Ser242 Cα.

I Omit 2Fo-Fc electron density for the R199H mutant.

J X-ray structure of the R199H mutant (blue) overlaid upon wild-type APTX (orange). Arg199 forms a salt bridge with Asp269 and a cation-π interaction with C-terminal residue Trp340. The arginine to histidine mutation causes the loss of these contacts resulting in an altered protein surface structure with a 2.1 Å shift from the protein core.

K Surface representation of APTX RNA-DNA complex showing both S242N and R199H mapped on a putative protein binding surface.

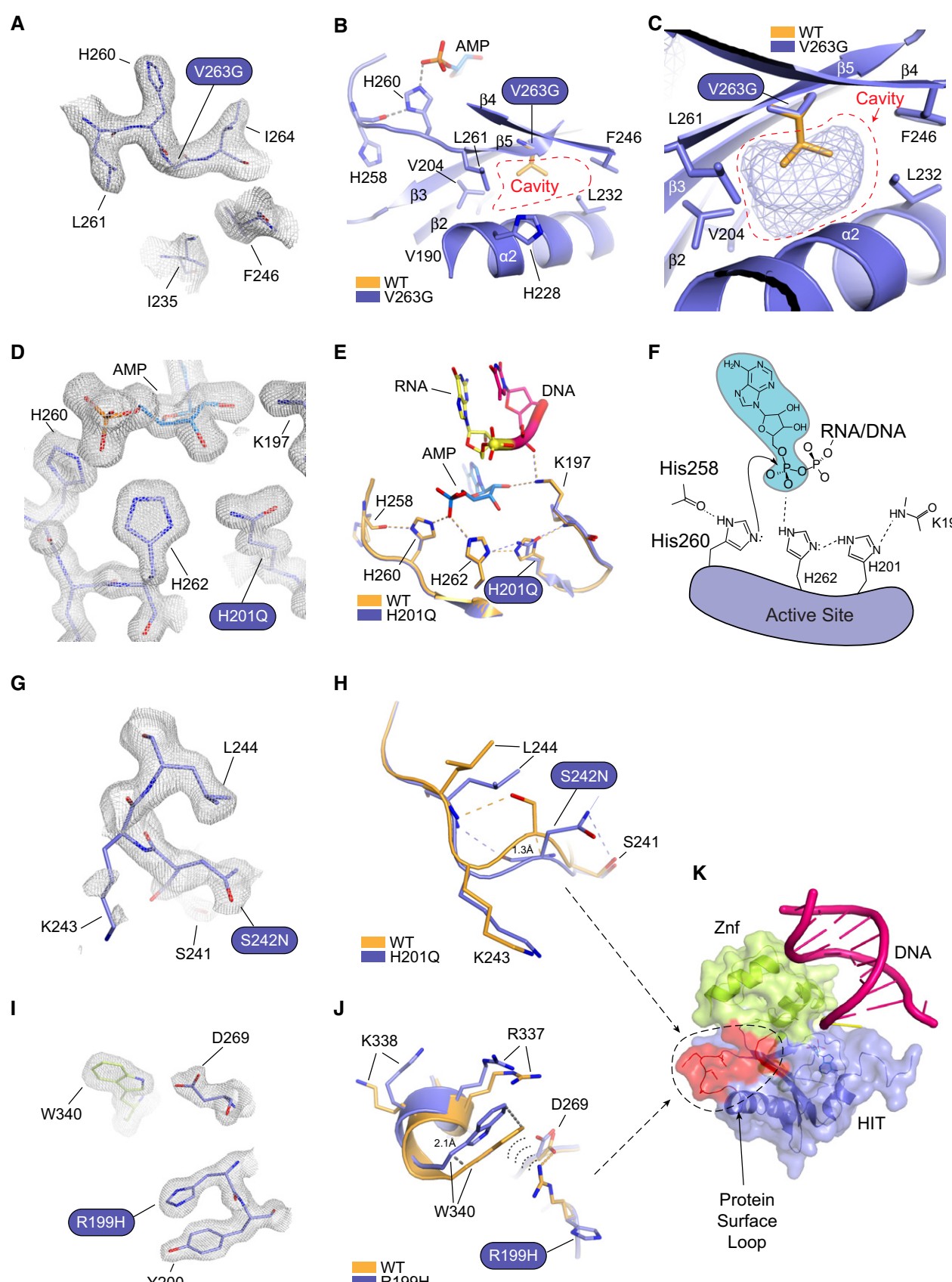

**Figure 4.**

phosphohydrolases, including APTX homologs. Based on the structure of an adenosine-vanadate transition-like state mimic, it is proposed that His201 enables stabilization of the negatively charged transition state of the deadenylation reaction (Tumbale *et al*, 2014; Schellenberg *et al*, 2015) via its participation in a charge relay consisting of the main-chain backbone of Lys197, His201, and His262 that hydrogen bonds to the scissile phosphate of the 5′-AMP phosphoanhydride (Fig 4D–F). The high-resolution structure of H201Q (at 1.65 Å) reveals the effects of this substitution on active site chemistry from the histidine to glutamine mutation (Fig 4D and E). In the structure of the H201Q mutant, although the active site undergoes minimal gross structural rearrangements, this charge relay is compromised, likely resulting in the significant reduction in catalytic activity (Fig 3D, Appendix Fig S6A and C) and protein stability (Fig 3C, and Appendix Fig S6C). Intriguingly, while seemingly well accommodated in the active site, the H201Q substitution also confers substantial temperature-dependent effects on catalysis. Close analysis of the backbone conformations of the Tyr200-Gln201 peptide bond reveals that a strained, but permissible deviation of peptide bond planarity (172–173°) is observed compared to wild-type APTX (Tumbale *et al*, 2014). This non-native backbone conformation in part could explain the thermal stability defect for H201Q.

*S242N*: Ser242 maps to a surface loop linking α2 and β4. The native loop conformation is stabilized by a hydrogen bond between the Ser242 hydroxyl and Leu244 main-chain peptide amide (Fig 4G and H). However, in the structure of the S242N mutant the Asn242 side-chain hydrogen bonds to Ser241 instead. These contacts stabilize rearrangements and an altered loop structure with a 1.3 Å shift at the Ser242-Cα between the wild-type and mutant structures. Overall impacts in protein folding are mild ($\Delta T_m = 3.5$°C) (Appendix Fig S6C), consistent with a surface loop destabilization of the structure.

*R199H*: Arg199 forms a salt bridge with Asp269, as well as participates in cation-π interactions with Trp340 (Fig 4I and J). The structure of the mutant reveals these interactions are lost in R199H. Trp340 shifts away from the protein core by 2.1 Å and the surrounding region of the Znf (α6) shifts ~0.5 Å away from the HIT domain. This suggests R199H destabilizes the HIT-Znf domain interface and explains the observed APTX stability ($\Delta T_m = -6.7$°C) and catalytic (14- to 18-fold reduced) defects. Intriguingly, the effect of this mutation propagates up into the region affected by S242N (α2–β4), and Ser242 makes an additional hydrogen bond to the carbonyl of Ala239. Thus, we hypothesize that by altering a common region of APTX these mutations may also affect a putative protein binding surface of currently unknown function (Fig 4K).

*L248M*: The dominant L248M APTX mutant was identified in AOA1 patients with severe clinical phenotype presenting early disease onset and a progressive ataxic syndrome with cerebellar atrophy, mental retardation, and epilepsy (Table 1) (Castellotti *et al*, 2011). This substitution is conservative and slightly stabilizing in the presence of AMP (Fig 3C), yet unexpectedly has a ~7-fold impact on APTX deadenylation at 37°C (Fig 3D, Appendix Fig S6C). Intriguingly, the crystal structure of L248M shows limited overall conformational perturbation of the DNA bound protein, with structural changes limited to minor shifts in neighboring His228 (Fig 5A). However, Leu248 maps to strand β4, is found in the protein hydrophobic core, and flanks regions of the protein undergoing

conformational changes between the catalytic domain closed and opened states (Appendix Fig S2). These conformational changes extend from α1 to residues surrounding the active site and Leu248 also participates in this network. Thus, we hypothesized that L248M might impact solution conformational states of the protein.

Consistent with this hypothesis, and in stark contrast to the DNA bound X-ray structure of the L248M protein, apo L248M displayed significant global conformational perturbations in $^{1}$H-$^{15}$N HSQC spectra of unliganded protein compared to WT (Fig 6B, Appendix Fig S8). The mutant impacts residue resonances that are > 16 Å from the mutation site. Similarly, the $^{1}$H-$^{13}$C-Met HSQC spectra showed non-native chemical shifts for L248M in a subset of both the Apo and reaction product bound states (Fig 5C). However, these differences were less apparent than those observed for WT versus L248M $^{1}$H-$^{15}$N samples (Appendix Fig S8 and Fig 5B). By comparison, the transition-like state (adenosine-vanadate reacted) displayed very similar $^{1}$H-$^{13}$C HSQC-Met spectra for WT and L248M APTX (Fig 5C, middle panel), consistent with the mutant protein being trapped in a catalytically competent state. That is, while apo and product states are non-native, covalent trapping of the transition state might mitigate these conformational differences. Given the distance from the catalytic center (> 10 Å), we assert that L248M substitution impacts active site assembly allosterically, possibly through a network of interactions within the protein core that are influenced by this substitution (Fig 5B).

## Discussion

Our APTX nicked RNA-DNA complex X-ray structure, NMR analysis of solution APTX-DNA binding properties, and structural and biochemical characterization of AOA1 mutants provided us with three important new insights into APTX structure and inactivation in disease. First, we have illuminated the conserved determinants of APTX nicked DNA engagement that facilitate the APTX deadenylation direct-reversal DNA repair reaction. Second, the APTX catalytic core is dynamic and regulated by substrate engagement. Structural rearrangements involve the amino-terminal DNA nick sensing helical wedge and reorganization of the APTX active site, indicating that APTX employs a DNA damage-induced fit mechanism for substrate recognition and active site assembly. Third, this work defines a detailed framework for understanding and categorizing the impacts of APTX catalytic domain missense and nonsense substitutions linked to AOA1 neurological disease.

The APTX double-barbed wedge segregates the upstream and downstream halves of the DNA damage site to facilitate a "wedge-pivot-cut"-mediated DNA damage direct-reversal reaction. 5′-AMP strand binding is fortified through the HIT-Znf composite DNA binding interface. Data from biochemical analysis of the R306X AOA1 truncation variant support the conclusion that two-point engagement of the lesion bearing strand enables efficient deadenylation repair. At the damaged DNA nick juncture, the aromatic histidine and tryptophan rings of the APTX α1 wedge induce DNA splaying with a ~90° bend. Such helical wedging provides a pivot point for the redirection of 5′ and 3′ strands, and this mechanism emerges as a common theme in DNA nick recognition. Notable

**Table 1.   Impacts of AOA1 mutations.**

| Mutants | Allele | Classes | Solubility | Predicted mutagenic effects on APTX structure and function | Mean age at onset (year of age) | Disease severity | References |
|---|---|---|---|---|---|---|---|
| A198V | A198V/A198V; A198V/P206L | Highly destabilizing | − | The larger valine side chain causes steric clash, disrupting bonding interactions at HIT and Znf domain interface. | 5 | XX | Le Ber *et al* (2003), Criscuolo *et al* (2004) |
| P206L | P206L/P206L; A198V/P206L | Highly destabilizing | − | The larger leucine side chain causes steric clash, disrupting hydrophobic interactions. | 11 | XX | Date *et al* (2001), Criscuolo *et al* (2004), Castellotti *et al* (2011), Moreira *et al* (2001) |
| G231E | G231E/689insT | Highly destabilizing | − | The larger and charged glutamate side chain causes steric clash, disrupting hydrophobic interactions. | 1 | XXX | Ito *et al* (2005) |
| R247X | R247X/R247X | Highly destabilizing | − | Truncation at R247 causes loss of the entire Znf domain. | n.r. | n.r. | Mosesso *et al* (2005) |
| W279X | W279X/W279X; D267G/W279X; W279X/R306X; W279X/Q181X; W279X/I159 fs | Highly destabilizing | − | Truncation at W279 causes loss of the entire Znf domain. | 5 | XX | Le Ber *et al* (2003), Castellotti *et al* (2011), Barbot *et al* (2001), Moreira *et al* (2001) |
| H201R | H201R/H201R | Highly destabilizing | − | The larger arginine side chain causes steric clash in the active site, resulting in protein unfolding. | 7.5 | XX | Shimazaki *et al* (2002) |
| H201Q | H201Q/H201Q | Moderately destabilizing | + | Distortion of active site architecture and chemistry. | 29 | X | Criscuolo *et al* (2004) |
| D185E | D185E/WT | Moderately destabilizing | + | The larger glutamate side chain disrupts H-bonding contacts and causes steric clash. | 18 | X | Castellotti *et al* (2011) |
| L223P | L223P/L223P | Moderately destabilizing | ++ | The leucine to proline mutation causes kinking in α2, altering protein surface. | 40 | X | Criscuolo *et al* (2005) |
| W279R | W279R/W279R; W279R/IVS5 + 1 | Moderately destabilizing | ++ | The larger and charged arginine side chain disrupts hydrophobic contacts, causing protein instability. | 5 | XX | Le Ber *et al* (2003), Castellotti *et al* (2011) |
| D267G | D267G/W279X | Moderately destabilizing | + | The smaller and uncharged glycine side chain causes loss of bonding contacts at the HIT and Znf domain interface. | 16.5 | XX | Le Ber *et al* (2003) |
| V263G | V263G/V263G; V263G/P206L | Moderately destabilizing | + | The smaller glycine side chain creates a cavity in protein core. | 19–25[a] | X | Date *et al* (2001), Yokoseki *et al* (2011) |
| K197Q | K197Q/W279X | Mildly destabilizing | +++ | The glutamine side chain causes loss of contacts with DNA phosphate and AMP ribose, impairing DNA binding. | 15 | XX | Tranchant *et al* (2003) |
| R199H | R199H/? | Mildly destabilizing | +++ | The arginine to histidine mutation results in altered putative protein binding surface. | 2.5 | XX | Barbot *et al* (2001), Moreira *et al* (2001) |
| S242N | S242N/? | Mildly destabilizing | +++ | The serine to asparagine mutation results in rearrangement of side chain contacts, altering surface loop structure. | 38 | X | Baba *et al* (2007) |
| R306X | R306X/R306X; R306X/W279X | Mildly destabilizing | +++ | Truncation at R306 causes partial loss of Znf domain. | 6.5 | XX | Castellotti *et al* (2011) |
| L248M | L248M/WT | Stabilizing | +++ | The leucine to methionine mutation hinders allosteric regulation of active site assembly coupled to catalysis. | 1–16[a] | XX-XXX | Castellotti *et al* (2011) |

+++ = comparable to WT, ++ = moderately soluble, + = mildly soluble, − = insoluble, XXX = age onset < 2 years of age, XX = age onset > 2–18 years of age, X = age onset > 18 years of age, n.r. = not reported.
[a]Exact age not reported.

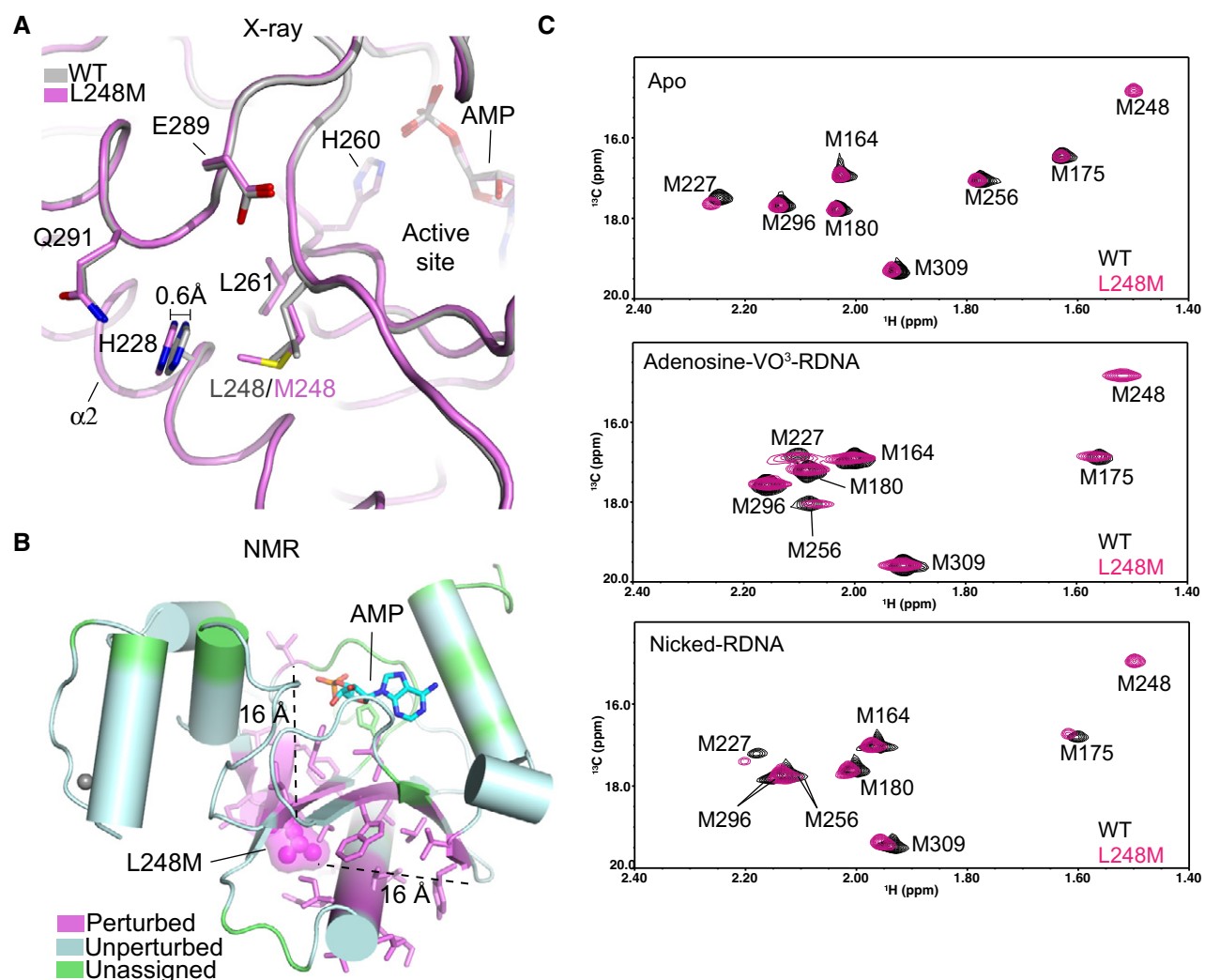

**Figure 5. X-ray crystallographic and NMR analyses of L248M.**

A  X-ray structural overlays of AOA1 mutant L248M (purple) and wild-type APTX (gray).

B  Location of residues exhibiting $^1$H-$^{15}$N NMR chemical shift perturbation (also see Appendix Fig S8) due to the L248M mutation. Perturbed (magenta), unperturbed (cyan), unassigned (green).

C  Assigned $^1$H-$^{13}$C HSQC spectral overlay of [methyl-$^{13}$C]methionine-labeled wild-type APTX and L248M mutant in APO (top), transition state (middle), and product bound state (bottom).

similar examples include DNA polymerase β (POLβ) and FEN1 nucleases, both of which provide alternate processing pathways for 5′-AMP repair (Daley *et al*, 2010; Caglayan & Wilson, 2017; Uson *et al*, 2017). Strikingly, these enzymes catalyze diverse reactions: (i) direct reversal of adenylation (APTX), (ii) AP lyase removal of damaged termini, in the case of abortive ligation on 5′-deoxyribose phosphate (5′-dRP) nicks (POLβ), or (iii) endonucleolytic incision of the damaged 5′ strand (FEN1) (Fig 6A) (Tsutakawa *et al*, 2011). These convergent solutions to resolving adenylated 5′-DNA ends all involve wedge-pivot-cut strategies built upon diverse domain folds and chemistry for removal of bulky adenylated termini (Fig 6B–D). Given that 3′ DNA damage can initiate the abortive ligation cycle (Harris *et al*, 2009; Daley *et al*, 2010; Schellenberg *et al*, 2015), we speculate APTX-DNA wedging might also permit access to the 3′ terminus for concerted or

subsequent action of DNA-end repair enzymes such as polynucleotide kinase/phosphatase (Bernstein *et al*, 2005) for processing of 3′ damage flanking the adenylated lesion from abortive DNA ligase action.

Conformational changes and disorder-to-order transitions of the wedge helix regulate assembly of the APTX active site lid. Based on the observed structural states, NMR spectroscopy and protease protection results, the DNA wedge helix adopts at least three states: (i) a disordered ligand free state, (ii) an α-helical DNA bound precatalytic (opened) state, and (iii) a closed α-helical catalytically competent conformer with an aligned APTX HIT active site loop. This mobile catalytic framework is further impacted by APTX mutations. The conformational changes observed in our structures extend deep into the protein core. Strikingly, hydrophobic core residues undergoing conformational changes in our structures are flanked by

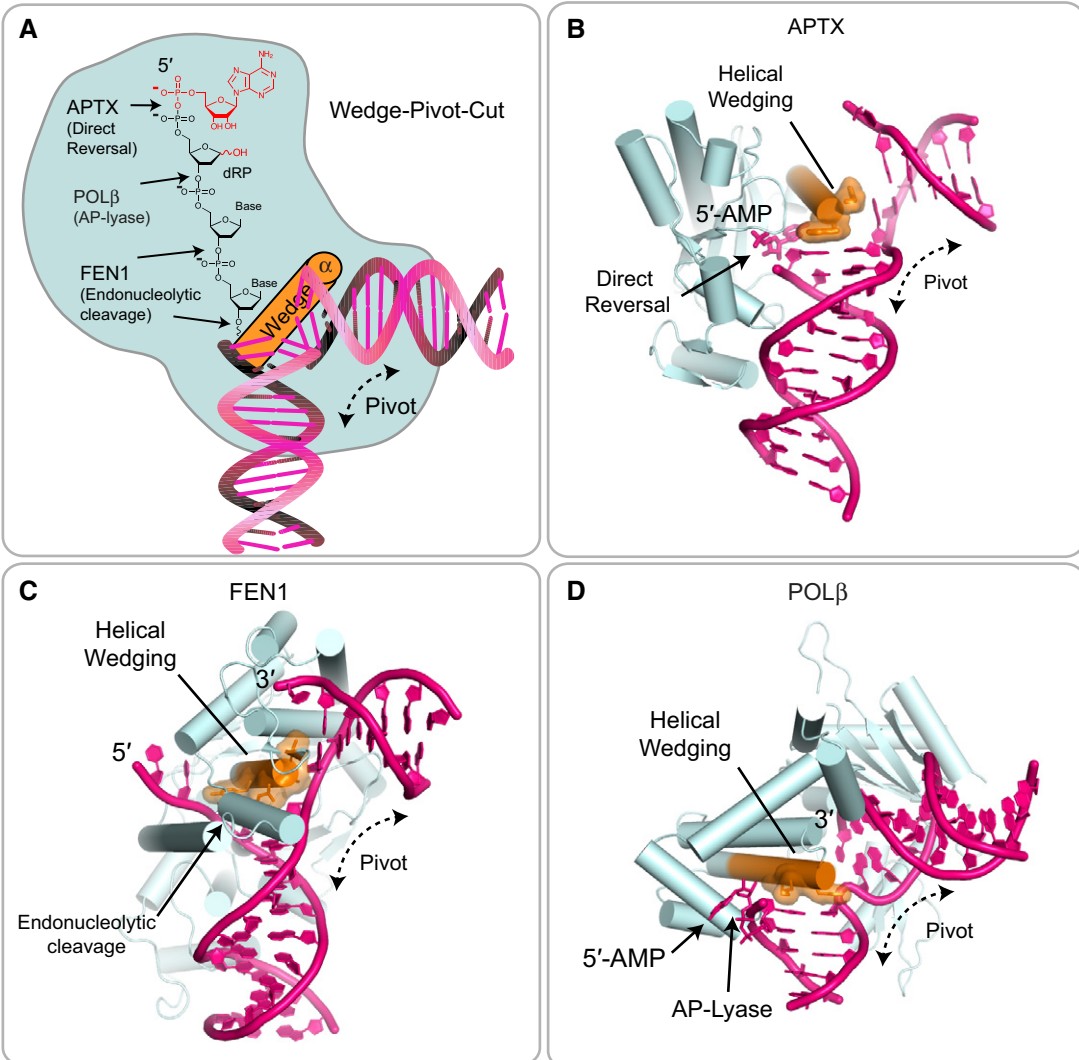

**Figure 6.  A wedge-pivot-cut mechanism underlies alternate 5′-AMP processing pathways.**

A   The APTX α1 wedges into the damaged nick, inducing a ~90° bend in the DNA and exposing both the 3′ and 5′ termini. In addition to APTX, DNA polymerase β (POLβ) and FEN1 nucleases provide alternate processing pathways for 5′-AMP repair.

B   Direct reversal of adenylation by APTX.

C   Endonucleolytic incision of the damaged 5′ adenylated strand by FEN1.

D   Removal of adenylated 5′-deoxyribose phosphate (5′-dRP) by POLβ.

residues mutated in AOA1, including the beta sheet residue Leu248 (Castellotti *et al*, 2011). NMR results show the L248M mutation impacts a network of surrounding secondary structure, rendering the domain "arthritic", such that DNA binding-induced conformational changes are impaired. The Leu248 side chain on β4 extends toward Gly231 on helix α2. Glycine is generally considered to be an unfavorable residue for α-helix stability (Malkov *et al*, 2008), but its position opposite Leu248 on β4 indicates that it may reinforce correct alignment of β4 with α2. The presence of the smaller Gly side chain paired with larger opposing side chains in adjacent structural elements has been suggested to play similar roles in other structures (Eilers *et al*, 2002; Liu *et al*, 2005; Mueller *et al*, 2005). In APTX, this arrangement allows Leu248 to intercalate between α2-helix residues His228 and Ile235. Extension of the Met248 methyl group into the

face of His228 is primarily responsible for the extreme upfield $^1$H shift of the methyl resonance (Fig 5C).

Overall, the solubility, protein stability, and activity assays broadly segregate AOA1 mutations into four protein stability classes (Table 1, Fig 7) defined here as follows: (i) highly destabilizing insoluble mutants (R247X, W279X, A198V, H201R, P206L, and G231E), (ii) moderately destabilizing mutations with $T_m$-Apo and $T_m$-AMP < 40°C (D185E, W279R, L223P, V263G, D267G, and H201Q), (iii) mildly destabilizing with $T_m$-Apo and $T_m$-AMP > 40°C (K197Q, S242N, R199H, and R306X), and (iv) stabilizing in the presence of ligand (L248M). Our biophysical results provide an atomic level rationale for instability of APTX mutant protein variants in AOA1 patient mutation derived cell lines that has been reported for several of the variants (W279X, P206L, V263G, R306X) (Gueven

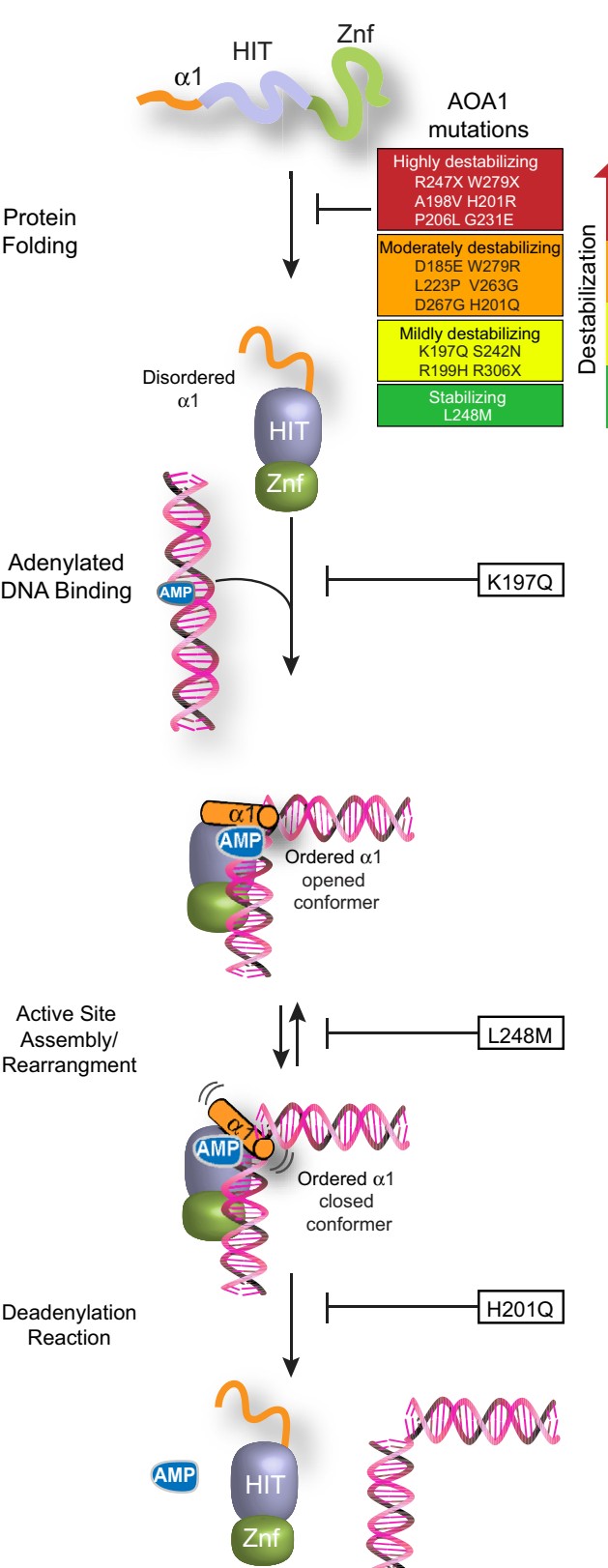

**Figure 7. Molecular impacts of AOA1 mutations on APTX structure and activity.**

AOA1 mutations are broadly classified into four classes based on their thermal stability: (i) highly destabilizing, (ii) moderately destabilizing, (iii) mildly destabilizing, and (iv) stabilizing. The highly destabilizing mutations cause protein unfolding. Other mutations reduce protein stability. AOA1 mutations have deferential impacts on APTX activity. The K197Q mutation causes DNA binding defects, L248M impairs active site assembly, and H201Q abolishes active site chemistry.

three mutants that impact DNA binding (K197Q), active site assembly/conformational changes (L248M), and deadenylation reaction and active site chemistry (H201Q) can be ascribed (Fig 7). Previous work showed that the K197Q mutation directly influences the substrate binding pocket (Tumbale *et al*, 2014), consistent with the impact on DNA binding-induced ordering of the wedge helix (Fig 2B). The stabilizing effect of L248M and its allosteric impact on catalytic domain conformations suggest that this mutation influences active site assembly and disassembly. Precisely how the H201Q substitution disrupts the APTX charge relay system will require future NMR or neutron scattering analysis of active site protonation states.

A survey of age of onset and disease severity does not yield clear correlates with protein stability or catalytic activity (Table 1). This suggests that additional genetic and/or environmental factors influence disease onset and progression. The progressive nature of AOA1 is typified by complex gene–environment interactions rooted in both mitochondrial and nuclear DNA repair defects conferred by *APTX* deficiency (Sykora *et al*, 2011; Akbari *et al*, 2015; Schellenberg *et al*, 2015). The subtle but ultimately catastrophic protein alterations defined herein likely trigger a cascade of molecular, cytological, and tissue-level degenerative developments in a time span of years to decades. Individual outcomes are also likely to be context dependent and may be impacted by the extent to which other repair pathways such as FEN1-dependent excision (Daley *et al*, 2010) are able to compensate for the loss of APTX activity. It has also been demonstrated that the base excision DNA repair proteins PARP-1, apurinic endonuclease 1 (APE1), and OGG1 are expressed at reduced levels in AOA1 cells concomitant with loss of APTX protein (Harris *et al*, 2009). Thus, a complex interplay between APTX deficiency and the modulation of base excision repair BER capacity in AOA1 also exists. Together, the results provide a mechanism for APTX deficiencies in AOA1 that may directly be linked to APTX catalytic deficiency and instability, and/or to more complex alterations in the APTX protein interactome that includes key base excision repair factors.

Our integrated APTX functional and structural studies provide testable structural paradigms for DNA-nick and DNA-end cleansing enzymes in the DNA damage response pathways, and better establish a molecular platform for understanding APTX dysfunction in neurodegenerative disease. Though AOA1 is a rare disorder, outcomes of this research have broad implications for understanding modes of APTX mutagenic inactivation in disease and individual genetic susceptibility to environmentally induced DNA damage. Micro-RNA modulation of APTX expression can radio-sensitize tumor cells (Wang *et al*, 2016). Thus, rational design of inhibitors that modulate APTX-DNA-end and DNA-nick deadenylation activities stemming from this work could be useful for treatment of cancers, or as a co-therapy with existing

*et al*, 2004; Hirano *et al*, 2007; Castellotti *et al*, 2011) studied here and suggest that APTX protein destabilization is a common mode of inactivation in AOA1. Specific added functional defects for

chemotherapeutics that induce "dirty" complex structured DNA breaks.

## Materials and Methods

### Cloning, mutagenesis, protein expression, and purification

His-tagged human APTX (hAPTX) protein variants were expressed and purified as previously described with some modifications (Tumbale *et al*, 2014). hAPTX variants were generated using Quick-Change site-direct mutagenesis kit (Stratagene). hAPTX wild-type and variants were expressed as N-terminal His-tagged proteins in *E. coli* Rosetta 2 (DE3) cells (Novagen). Cell cultures were grown at 37°C in LB medium supplemented with ampicillin (100 μg/ml) and chloramphenicol (34 ng/ml) until $A_{600}$ reached 1, at which 50 μM IPTG was added to cell cultures. Protein expression was carried out at 16°C overnight. Cells were harvested by centrifugation (16,000 $g$, 10 min). Cell pellet was resuspended and lysed in 30 ml (50 mM Tris, pH 8.5, 500 mM NaCl, 10 mM imidazole, 0.1 g lysozyme/1l pellet, 1 tablet Roche mini EDTA-free protease inhibitor cocktail) and incubated at 4°C for 30 min, followed by sonication. The soluble fraction was applied to Ni-NTA resins (5 ml packed volume) (Qiagen) and washed with 100 ml (50 mM Tris, pH 8.5, 500 mM NaCl, 10 mM imidazole), 15 ml (50 mM Tris, pH 8.5, 500 mM NaCl, 30 mM imidazole). The His-tagged protein was eluted in 15 ml (50 mM Tris, pH 7.5, 500 mM NaCl, 300 mM imidazole). The His-tag was removed by thrombin cleavage (50 U) (Sigma) at 4°C overnight. The untagged protein was purified on a Superdex 75 gel filtration column (GE healthcare) in gel filtration buffer (50 mM Tris, pH 7.5, 500 mM NaCl, 5% glycerol, 0.1% βmercaptoethanol), followed by cation exchange chromatography on HiTrap SP HP 5 ml column (GE healthcare). The purified proteins were analyzed by SDS–PAGE and stored in 25 mM Tris, 150 mM NaCl, 5% glycerol at −80°C until use.

### $^{15}$N- and $^{13}$C-methionine labeling hAPTX

The $^{13}$C-methionine-labeled hAPTX proteins were expressed in *E. coli* Rosetta 2 (DE3) cells in M9 growth medium containing ampicillin (100 μg/ml) and chloramphenicol (34 ng/ml) (supplemented with 0.5% glucose, 2 mM MgSO4, 1,000× trace metals (200 μl/l), 20 amino acids (except for proline, alanine and glycine) (200 mg/l), 200 mg/l $^{13}$C-methyl methionine. The $^{15}$N-labeled proteins were expressed in M9 medium containing $^{15}$NH$_4$Cl with ampicillin (100 μg/ml) and chloramphenicol (34 ng/ml). Protein expression and purification methods were as described for unlabeled APTX variants above.

### NMR assignments

Methionine Cε and Hε resonances were assigned by creating site directed M to L mutations of each methionine residue and comparing the $^{13}$C-$^1$H HSQC spectra of $^{13}$C methyl methionine-labeled mutants with the wild-type hAPTX spectra. Assignments of the backbone $^1$H, $^{13}$C, and $^{15}$N resonance were achieved with standard techniques and are nearly identical to that reported by Bellstedt *et al* (2013), and are reported in the BMRB, accession 27287. Notably, the APTX

constructs used here (aa 151–307) did not suffer the precipitation problems previously encountered (Bellstedt *et al*, 2013).

### hAPTX AOA1 variants solubility assays

hAPTX$^{cat}$ WT and variants were grown in 3 ml LB medium containing ampicillin (100 μg/ml) and chloramphenicol (34 ng/ml) at 37°C. Cell cultures were induced with 100 μM IPTG when $A_{600}$ = 1. Protein expression was carried out at 16°C overnight. Cell density was normalized to $A_{600}$ = 1. Cell pellet from 1 ml cell culture was suspended and lysed in 200 μl (50 mM Tris, pH 8.0, 100 mM NaCl, 5 mM EDTA, 0.25 mg/ml lysozyme, 10 μg/ml DNaseI, 5 mM MgCl$_2$) at 4° for 1 h. Cell lysate was fractionated by centrifugation (30,000 $g$, 10 min). Whole cell lysates (5 μl) and soluble fractions (5 μl) were boiled in 2× SDS sample buffer (5 μl) for 10 min, followed by centrifugation (30,000 $g$, 10 min). Western blotting for his-tagged hAPTX$^{cat}$ WT and variant proteins in whole cell lysate and soluble fractions used HRP mouse anti-His$_6$ antibodies (1:5,000) (BD Pharmingen), followed by HRP-conjugated rabbit anti-mouse secondary antibodies (1:5,000). Membranes were treated by ECL detection kit (GE Healthcare) and then exposed to chemiluminescence film (GE Healthcare).

### hAPTX deadenylation reactions

Reaction mixtures (10 μl) contained 5′-AMP$^{RNA-DNA}$ (Tumbale *et al*, 2014) (10 nM) and hAPTX (2 nM) in 50 mM Tris, pH 7.5, 40 mM NaCl, 5 mM EDTA, 1 mM DTT, and 5% glycerol. Reactions were incubated at 25 or 37°C for 10 min, followed by inactivation at 97°C in 8 M urea, 50 mM Tris, pH 7.5, 25 mM EDTA, and 5% glycerol (10 μl) for 10 min. Denatured DNA was analyzed on 15% TBE-Urea gel (Invitrogen). Fluorescent-labeled reaction products were detected with a Typhoon scanner and quantified using ImageQuant (GE Healthcare).

### Limited proteolysis

Proteolysis reaction mixtures (60 μl) contained 22 μM full-length hAPTX WT or K197Q, ligands (2 mM AMP), blunt RNA/DNA (oligos 2 and 3, Appendix Table S3, 200 μM), nicked DNA (oligo 1, Appendix Table S3, 200 μM), plasmid DNA (1 mg), adenosine (2 mM), or vanadate (2 mM), and 2 μg chymotrypsin in 20 mM Tris, pH 7.5, 450 mM NaCl, and 0.1% β-mercaptoethanol. A 10 μl aliquot of the reaction mixture was removed at 5, 15, 30, and 60 min, immediately mixed with 1 μl 1 mg/ml PMSF, and heat inactivated in Novex SDS sample buffer (Invitrogen) at 95°C for 10 min. Proteolytic digestions were analyzed by SDS–PAGE, and protein was stained with Coomassie Blue.

### Thermal shift assay

Reaction mixtures (20 μl) contained hAptx (200 μg/ml), AMP (1 mM) or DMSO (1% v/v), Sypro orange dye (1:2,500) (Invitrogen) in 10 mM Tris, pH 7.5, 150 mM NaCl. Fluorescent intensity was collected starting at 25°C up to 95°C (2°C increment/min) with a QPCR machine using excitation and emission wavelengths of 492 nm and 610 nm, respectively (Niesen *et al*, 2007). Delta $T_m$ for a ligand is calculated as average $T_m$ (for controls) minus the observed $T_m$ for the protein in the presence of a ligand.

## hAptx-nicked DNA complex crystallization and structure determination

Crystals of the hAPTX-nicked-RNA-DNA–AMP–$Zn^{2+}$ complex (oligos 4 and 5, Appendix Table S3) were grown by mixing 300 nL of complex solution (10 mg/ml hAptx (165–342), 1 mM AMP, 1.5:1 DNA:protein molar ratio, in 150 mM NaCl, 20 mM Tris–HCl, pH 7.5, and 0.1% β-mercaptoethanol) with an equal volume of precipitant solution (100 mM MES, pH 6.5, 16% (w/v) polyethylene glycol 3350) at 4°C. X–ray diffraction data were collected on beamline 22-ID of the Advanced Photon Source at a wavelength of 1,000 Å. X-ray diffraction data were processed and scaled using the HKL2000 suite (Otwinowski & Minor, 1997). The hAPTX–nicked-RNA-DNA–AMP–$Zn^{2+}$ complex structure was solved by molecular replacement using PDB entry 4NDG (Tumbale *et al*, 2014) as a search model with PHASER (McCoy *et al*, 2007) executed in the PHENIX software suite (Adams *et al*, 2011). Iterative rounds of model building in COOT (Emsley *et al*, 2010) and refinement with PHENIX were used to produce the final models that have excellent geometry (Appendix Table S2).

The crystallographic asymmetric unit contains one doubly nicked DNA substrate and two APTX protomers (Fig 1B and C). At the nick junctions, DNA base intercalation unwinds both of the 5′ and 3′ base pairs proximal to the nick site. These DNA bending and unwinding events are fundamentally different in the two bound complexes in the crystallographic asymmetric unit (Appendix Fig S1B). The local unwinding events at the nick junctions are also stabilized in the crystal by a series of microhomology-mediated pairing events between the two non-crystallographic related molecules. The position of the upstream region of the nick in each complex also differs. While the upstream regions are bound orthogonally relative to the downstream duplex (8 bp region), structural overlays shows the nick can be positioned in alternate binding modes related by an ~ 90° degree relative rotation to one another (Appendix Fig S1B).

## Mutant hAPTX Crystallization and structure determination

Crystals of the hAPTX-R199H/RNA-DNA/AMP-$Zn^{2+}$ product complex (Oligos 2 and 3, Appendix Table S3) were grown by mixing 300 nl of complex solution (10 mg/ml hAPTX-R199H (165–342), 1 mM AMP, 1.5:1 RNA/DNA:protein molar ratio, in 150 mM NaCl, 20 mM Tris–HCl, pH 7.5, and 0.1% β-mercaptoethanol) with an equal volume of precipitant solution (100 mM MES, pH 6.5, 20% (w/v) polyethylene glycol 3350) at 4°C. Crystals of the hAPTX-H201Q/RNA-DNA/AMP-$Zn^{2+}$ product complex (Oligos 2 and 3, Appendix Table S3) were grown by mixing 300 nl of complex solution (9 mg/ml hAPTX-H201Q (165–342), 1 mM AMP, 1.5:1 RNA/DNA:protein molar ratio, in 150 mM NaCl, 20 mM Tris–HCl, pH 7.5, and 0.1% β-mercaptoethanol) with an equal volume of precipitant solution (100 mM sodium HEPES, pH 7.5, 25% (w/v) polyethylene glycol 3350) at 4°C. Crystals of the hAPTX-S242N/RNA-DNA/AMP-$Zn^{2+}$ product complex (Oligos 2 and 3, Appendix Table S3) were grown by mixing 300 nl of complex solution (10 mg/ml hAPTX-S242N (165–342), 1 mM AMP, 1.5:1 RNA/DNA:protein molar ratio, in 150 mM NaCl, 20 mM Tris–HCl, pH 7.5, and 0.1% β-mercaptoethanol) with an equal volume of precipitant solution (200 mM ammonium formate, 20% (w/v) polyethylene glycol 3350) at 4°C. Crystals of the hAPTX-L248M/DNA/AMP-$Zn^{2+}$ product complex (Oligo 6, Appendix Table S3) were grown by mixing 300 nl of complex solution (10 mg/ml hAPTX-L248M (165–342), 1 mM AMP, 1.5:1 DNA:protein ratio, in 150 mM NaCl, 20 mM Tris–HCl, pH 7.5, and 0.1% β-mercaptoethanol) with an equal volume of precipitant solution (100 mM MES, pH 6.5, 25% (w/v) polyethylene glycol 4000) at 4°C. Crystals of the hAPTX-V263G/RNA-DNA/AMP-$Zn^{2+}$ product complex (Oligos 2 and 3, Appendix Table S3) were grown by mixing 300 nl of complex solution (5 mg/ml hAPTX-V263G (165–342), 1 mM AMP, 1.5:1 RNA/DNA:protein molar ratio, in 150 mM NaCl, 20 mM Tris–HCl, pH 7.5, and 0.1% β-mercaptoethanol) with an equal volume of precipitant solution (100 mM MES pH 6.5, 20% (w/v) polyethylene glycol 3350). Crystals were washed in cryo-protectant (precipitant solution supplemented with 12% (v/v) glycerol) and flash frozen in liquid nitrogen for data collection.

## Data availability

The NMR data from this publication have been deposited in the Biological Magnetic Resonance Data Bank (http://www.bmrb.wisc.edu/) and assigned the identifier 27287.

Structural coordinates from X-ray crystallographic studies in this publication have been deposited in the RCSB protein data bank (https://www.rcsb.org/) and assigned the identifiers 6CVO, 6CVP, 6CVQ, 6CVR, 6CVS, and 6CVT (see Appendix Table S2).

**Expanded View** for this article is available online.

## Acknowledgements

Our studies are supported by the US National Institute of Health Intramural Program, US National Institute of Environmental Health Sciences (NIEHS), 1Z01ES102765 (to R.S.W.) and 1ZIA ES050111-26 (to R.E.L.), and Cancer Research UK (C480/A11411 to I.W). We thank L. Pedersen of the NIEHS Collaborative crystallography group for data collection support and the Advanced Photon Source (APS) Southeast Regional Collaborative Access Team (SER-CAT) for beamline access. Use of the APS was supported by the U. S. Department of Energy, Office of Science, Office of Basic Energy Sciences, under Contract No. W-31-109-Eng-38. We thank Dr. Bill Copeland and Dr. Monica Pillon for comments on the manuscript.

## Author contributions

Conceptualization RSW, PT, GAM, REL; methodology PT, GAM, JK, EF, MJS, JNL, MW, IW, REL RSW; investigation PT, MJS, GAM, EM, MW, JK, JNL, RSW; writing—original draft, RSW; writing—reviewing and editing, PT, MJS, GAM, JK, EF, JNL, MW, IW, REL, RSW; funding acquisition REL, IW, RSW; supervision REL, IW, RSW.

## Conflict of interest

The authors declare that they have no conflict of interest.

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
