## [Review Process File · The EMBO Journal]

Mechanism of APTX Nicked DNA Sensing and Pleiotropic Inactivation in Neurodegenerative Disease

Percy Tumbale, Matthew J. Schellenberg, Geoffrey A. Mueller, Emma Fairweather, Mandy Watson, Jessica N. Little, Juno Krahn, Ian Waddell, Robert E London, and R. Scott Williams

Review timeline:

Submission date:	19 December 2017
Editorial Decision:	5 February 2018
Revision received:	27 April 2018
Accepted:	29 May 2018

Editor: Hartmut Vodermaier

Transaction Report:

1st Editorial Decision

5 February 2018

Thank you for submitting your manuscript on nicked DNA sensing by APTX for our editorial consideration. After some delay related to the high number of submissions and limited reviewer availability at the turn of the years, we have now heard back from all three referees that had agreed to assess this work. In light of these comments, we shall be happy to consider this study further for publication, pending adequate revisions of a number of specific issues raised in the comments below. I would therefore like to invite you to prepare and upload a revised version of the manuscript, following the guidelines below and in our Guide to Authors. Please note that it is our policy to allow only a single round of major revision, thus making it important to carefully answer to all points raised at this stage. We might further discuss possible extension of the revision period (beyond the regular three months), during which time the publication of any competing work elsewhere would have no negative impact on our final assessment of your own study.

Thank you again for the opportunity to consider this work for The EMBO Journal, and please do not hesitate to get in contact should you have any further questions regarding the referee reports or this decision. I look forward to your revision.

Referee #1:

Williams and colleagues present an elegant systematic structural survey with an attempt to explain how APTX recognises single-strand breaks, as opposed to blunt ended double-strand breaks that has been reported, and how APTX mutations in AOA1 correlate with changes in structure, stability and activity. The authors report an induced fit mechanism of wild type APTX and provide some insight on how AOA1 variants impact APTX active site conformation and stability. The manuscript is well written and the experiments are carefully designed and executed. This is a hard-core structural biology paper and I am sure other reviewers will cover technical aspects that I am less familiar with.

I'd like however to make the following comments:

- 1- Nicked substrates are the most abundant physiological lesions of APTX. A strength of this study compared to previous work is uncovering the structure using this substrate and thus it'd be great if the authors can capitalise on this strength to look at AOA1 variants. Will using truncated proteins featuring the APTX mutations improve the protracted crystallisation time and render them more amenable to structural analyses?
- 2- The age of disease onset and severity do not seem to correlate with APTX stability or activity. What is the physiological relevance of the structural data? Do the mutants co-purify with other factors that makes them more or less detrimental on cell function? If addressing this experimentally is beyond the scope of this study, the authors may speculate a bit further on this discrepancy. What these factors might be?
- 3- I'd have liked to see some backing up of the in vitro data with cellular experiments, for example looking at the stability of AOA1 patient cells harbouring the studied mutations at conditions of hyperthermia. Do these variants interact in the cell with other factors that would render them more or less stable?
- 4- For the rational design of APTX inhibitors, it'd be important to compare and contrast between the mode of action on blunt ended DSBs versus nicked SSBs and whether this knowledge would inform or change the current thinking of designing inhibitors?

Referee #2:

In this manuscript, Tumbale and colleagues combined an array of approaches including X-ray crystallography, NMR spectroscopy and biochemistry to quantitatively probe how the polynucleotide deadenylase Aprataxin (APTX) processes adenylation damage in the context of RNA-DNA junctions. Mutations in the APTX gene have been linked to neurodegenerative disorder Ataxia with Oculomotor Ataxia 1 (AOA1). The crystallography data generated by these investigators provide convincing evidence for a DNA damage induced fit mechanism for APTX substrate recognition and assembly of the enzyme active site. A particularly exciting aspect of the study is the solution NMR spectroscopy characterization of APTX conformational responses during catalysis. Changes in chemical shifts of [methyl-¹³C]-labeled methionine NMR signals were used as probes to investigate changes in APTX in its apo state, bound to blunt-ended RNA-DNA substrate, nicked RNA-DNA substrate, adenosine-VO₃-RNA-DNA transition state mimic, and nick AMP and RNA-DNA product bound state. The results of the NMR characterizations are consistent with the catalytic core of APTX being dynamic and regulated by association with the substrate as was suggested from X-ray diffraction studies. The NMR-based approach presented in the manuscript will certainly be useful for characterizing the modes of action of other enzymes. Tumbale and colleagues also examined the effect of several AOA1-causing mutations on the solubility, stability and activity of APTX. Five X-ray structures of APTX variants in complex with RNA-DNA and AMP were notably determined. The results of this investigation help generate testable hypotheses for the impacts of AOA1-related mutations as nicely illustrated for the L248M variant. Overall, this is a high quality study suitable for publication in the EMBO Journal.

Minor point

Small typos in table 1: see column with disabling/destabilizing/stabilizing. Same typos in Figure 8.

Referee #3:

The authors present a manuscript that describe the molecular details for how the protein APTX detects and binds to RNA-DNA substrates; proposing a mechanism for how the active site of APTX is remodelled upon substrate binding to generate a catalytically competent active site. They also describe and catalogue the effects of several mutations on APTX that are linked with the humans disease AO1.

Overall the manuscript is of a very high standard, and in general, its findings are warranted and

adequately supported by experimental evidence. With some amendment, it should be accepted for publication in the EMBO Journal.

Please see "Major points" below, for specific items that require some additional clarification, or further experiments in order to fully support the authors' conclusions.

Major points

Page 8

>>Is the increase in T_m for the L248M mutant (Figure 3C) actually significant? To my eye it looks like the error bars overlap. How has the value for T_m been calculated? A representative example of the two denaturation profiles should be shown (WT and L248M). Because the importance ascribed to this mutation, it should also be verified and supported by a separate orthogonal assay (melting by CD or similar).

Figure 4

>>Figure 4 should be moved to supplementary material; as it is not sufficiently described / explained in the body of the main text. An expanded figure legend is also required, in order to provide sufficient explanation for the effect of each mutation on APTX to those readers without a structural biology background.

Page 10

"Similarly, the 1H-13C-Met HSQC spectra showed non-native chemical shifts for L248M in both the Apo and reaction product bound states (Figure 6C)"

>> The only residue with a non-native chemical shift in the HSQC spectra for the L248M protein compared to Apo(WT) appears to be M227 - and even that appears marginal. Please consider amending this statement.

Page 10

"Extended X3 trans conformational states"

The meaning of this statement will not be immediately transparent to all potential readers - an expanded explanation is required at this point.

Minor points

>> Mean +/- s.d. should be amended to read Mean +/- 1 s.d. in all figure legends.

Figure 2B

>>The asterisk annotating the chymotrypsin digests are difficult to see - and will be too small when the figure is reduced for publication. The pink and green asterisks in particular do not seem to always line up with a digested species. Some level of quantification (i.e. a histogram) for the appearance/disappearance of the C1 species would assist the reader.

Page 3

"Aptx mutants are sensitive to the 4NQO (Deshpande et al 2009)"

>> Remove the word 'the' from the above sentence.

Figure 2D / Figure 2E

>> Both panels from the methyl-met NMR experiments should be plotted on the same x-axis limits (1.4 - 2.3 ppm) and both axes labelled consistently (i.e. include 16 and 20 on y-axis)

Figure 3C and 3D

>>Data for the wild-type protein should always be presented as the first bar in the histogram. It would also aid the reader to plot the histograms with a consistent order (i.e. not rank order) to allow a simple comparison. Perhaps using the clustering of mutation types presented later in the manuscript?

Figure 3D

>>There are no error bars associated with this histogram. Is this a simple oversight? - as the figure legend indicates that three technical replicates were performed.

Page 9

"Similar to core cavitation mutants of BRCA1(Williams et al, 2003; Williams et al 2001; Williams et al 2004)"

>> Are all three self-citations fully justified / warranted here?

Supplementary Figure 4B

>> Typo in table header - "Differnence"

1st Revision - authors' response

27 April 2018

(begins on next page)

Response to reviewers

Overall all three reviewers were quite supportive of this work. We thank the three referees for their constructive and supportive comments. Please find reviewer comments "*blue italics*" and our responses to all critiques in black text.

Referee #1:

Williams and colleagues present an elegant systematic structural survey with an attempt to explain how APTX recognises single-strand breaks, as opposed to blunt ended double-strand breaks that has been reported, and how APTX mutations in AOA1 correlate with changes in structure, stability and activity. The authors report an induced fit mechanism of wild type APTX and provide some insight on how AOA1 variants impact APTX active site conformation and stability. The manuscript is well written and the experiments are carefully designed and executed. This is a hard-core structural biology paper and I am sure other reviewers will cover technical aspects that I am less familiar with. I'd like however to make the following comments:

We thank reviewer 1 for their overall supportive comments.

1- Nicked substrates are the most abundant physiological lesions of APTX. A strength of this study compared to previous work is uncovering the structure using this substrate and thus it'd be great if the authors can capitalise on this strength to look at AOA1 variants. Will using truncated proteins featuring the APTX mutations improve the protracted crystallisation time and render them more amenable to structural analyses?

Despite substantial efforts, we have been unable to crystallize any of the mutants in the nicked DNA crystal form. In the revised manuscript, we also highlight this hurdle in the text. As the referee notes, the new crystal structures advance our understanding of the molecular basis for APTX engagement of its cognate nicked DNA substrates. To facilitate definition of the molecular impacts of subtle missense substitutions that can alter just few atoms, the blunt DNA end complexes that rapidly crystallize and diffract to high resolution were of key utility and provide the only molecular definition of the consequences of these disease causing mutants reported to date.

2- The age of disease onset and severity do not seem to correlate with APTX stability or activity. What is the physiological relevance of the structural data?

We did note in the discussion that a direct correlation with age of onset of AOA1 does not emerge from the analysis of T_m and activity of AOA1 mutants. This is perhaps not surprising. The APTX missense mutations studied herein are strongly associated with AOA1, a debilitating progressive neurodegenerative disorder that typically renders affected individual's wheelchair-bound by adolescence. The molecular impacts of these disease associated alleles have, by and large, not been defined until now. It is the root molecular defects that we aimed to shed light onto here. Like any novel structural data, the data ARE physiologically significant simply as we previously lacked data defining how AOA1 variants alter protein structure, chemistry and DNA processing activities.

The progressive nature of AOA1 is typified by complex gene-environment interactions rooted in both mitochondrial and nuclear DNA repair defects conferred by APTX deficiency. The subtle, but ultimately catastrophic protein alterations defined herein likely trigger a cascade of molecular, cytological, and tissue-level degenerative developments in a timespan of years to decades. Individual outcomes are also likely to be context dependent and may be impacted by the extent to which other repair pathways such as FEN1-dependent excision are able to compensate for the loss of APTX activity. It is therefore perhaps not unexpected that a clear correlation does not emerge. This fact does not invalidate the relevance of the molecular structural data documented here. We note these complexities in the revised manuscript discussion.

Do the mutants co-purify with other factors that makes them more or less detrimental on cell function? If addressing this experimentally is beyond the scope of this study, the authors may speculate a bit further on this discrepancy. What these factors might be?

3- I'd have liked to see some backing up of the in vitro data with cellular experiments, for example looking at the

stability of AOA1 patient cells harbouring the studied mutations at conditions of hyperthermia. Do these variants interact in the cell with other factors that would render them more or less stable?

The referee raises important points for discussion in the revised manuscript, and we agree that a detailed cell biological analysis is beyond the scope of this work. Acquiring access to such patient cell lines would take several months in the protracted NIH regulatory structure, and/or establishment of new collaborations to augment the already deep protein structure and biophysical content of the existing manuscript. However, these types of experiments have been published for a subset of AOA1 variants. We now discuss this important published data from Lavin and Colleagues (Harris et al, 2009) who demonstrated PARP-1, apurinic endonuclease 1 (APE1) and OGG1 are expressed at reduced levels in AOA1 cells concomitant with loss of APTX protein. This work previously pointed to a complex interplay between APTX deficiency in modulating altered base excision repair BER capacity in AOA1. We now include these important facts in the discussion. We also explicitly note examples where APTX protein levels were documented in clinical studies. Together with published work, our results provide a mechanism relating APTX deficiencies in AOA1 that may directly be linked to APTX catalytic deficiency, and/or to more complex alternations in the APTX protein interactome that includes key base excision repair factors.

4- For the rational design of APTX inhibitors, it'd be important to compare and contrast between the mode of action on blunt ended DSBs versus nicked SSBs and whether this knowledge would inform or change the current thinking of designing inhibitors?

We report a kinetic comparison of nicked versus blunt end processing of these activities in Figure 2F. We demonstrate APTX has superior catalytic efficiency on nicked substrates. To date, there have been no reported APTX inhibitors. We agree that appropriate assays such as those reported in Figure 2F, will be important for implementation of inhibitor screens.

Referee #2:

In this manuscript, Tumbale and colleagues combined an array of approaches including X-ray crystallography, NMR spectroscopy and biochemistry to quantitatively probe how the polynucleotide deadenylase Aprataxin (APTX) processes adenylation damage in the context of RNA-DNA junctions. Mutations in the APTX gene have been linked to neurodegenerative disorder Ataxia with Oculomotor Ataxia 1 (AOA1). The crystallography data generated by these investigators provide convincing evidence for a DNA damage induced fit mechanism for APTX substrate recognition and assembly of the enzyme active site. A particularly exciting aspect of the study is the solution NMR spectroscopy characterization of APTX conformational responses during catalysis. Changes in chemical shifts of [methyl-13C]-labeled methionine NMR signals were used as probes to investigate changes in APTX in its apo state, bound to blunt-ended RNA-DNA substrate, nicked RNA-DNA substrate, adenosine-VO3-RNA-DNA transition state mimic, and nick AMP and RNA-DNA product bound state. The results of the NMR characterizations are consistent with the catalytic core of APTX being dynamic and regulated by association with the substrate as was suggested from X-ray diffraction studies. The NMR-based approach presented in the manuscript will certainly be useful for characterizing the modes of action of other enzymes. Tumbale and colleagues also examined the effect of several AOA1-causing mutations on the solubility, stability and activity of APTX. Five X-ray structures of APTX variants in complex with RNA-DNA and AMP were notably determined. The results of this investigation help generate testable hypotheses for the impacts of AOA1-related mutations as nicely illustrated for the L248M variant. Overall, this is a high quality study suitable for publication in the EMBO Journal.

We thank reviewer 2 for their positive comments.

Minor point

Small typos in table 1: see column with disablizing/destablizing/stabilizing. Same typos in Figure 8.

Thank you. Typos fixed.

Referee #3:

The authors present a manuscript that describe the molecular details for how the protein APTX detects and binds to RNA-DNA substrates; proposing a mechanism for how the active site of APTX is remodelled upon substrate binding to generate a catalytically competent active site. They also describe and catalogue the effects of several mutations on APTX that are linked with the humans disease AO1.

Overall the manuscript is of a very high standard, and in general, its findings are warranted and adequately supported by experimental evidence. With some amendment, it should be accepted for publication in the EMBO Journal.

Please see "Major points" below, for specific items that require some additional clarification, or further experiments in order to fully support the authors' conclusions.

Major points

Page 8

>>Is the increase in Tm for the L248M mutant (Figure 3C) actually significant? To my eye it looks like the error bars overlap. How has the value for Tm been calculated? A representative example of the two denaturation profiles should be shown (WT and L248M). Because the important ascribed to this mutation, it should also be verified and supported by a separate orthogonal assay (melting by CD or similar).

The measurements obtained are quite precise, the errors displayed are correct, and the p-values indicate significant differences for the values in question ($P < 0.05$). We now include the original thermal shift traces for all experiments in the supplementary material. We agree that an orthogonal assay with CD could be of utility. However, with our CD instrument, the measurements that we have made on other systems do not have near the precision of the thermal shift assay (SD ~ 0.5 degrees), and also lack the throughput capacity of the thermal shift assay. It is for this reason that we utilized the superior highly reproducible and precise thermal shift assay for our Tm analysis.

Figure 4

>>Figure 4 should be moved to supplementary material; as it is not sufficient described / explained in the body of the main text. An expanded figure legend is also required, in order to provide sufficient explanation for the effect of each mutation on APTX to those readers without a structural biology background.

We have moved Figure 4 to the supplementary materials, and renumbered the figures.

Page 10

"Similarly, the 1H-13C-Met HSQC spectra showed non-native chemical shifts for L248M in both the Apo and reaction product bound states (Figure 6C)"

>> The only residue with a non-native chemical shift in the HSQC spectra for the L248M protein compared to Apo(WT) appears to be M227 - and even that appears marginal. Please consider amending this statement.

We have edited this statement to: " Similarly, the 1H-13C-Met HSQC spectra showed non-native chemical shifts for L248M in a subset of both the Apo and reaction product bound states. However, these differences were less apparent than those observed for WT versus L248M 1H-15N samples (Supplementary Figure 8 and Figure 5B).

Page 10

"Extended X3 trans conformational states"

The meaning of this statement will not be immediately transparent to all potential readers - an expanded explanation is required at this point.

We agree and feel this statement is extraneous, and its removal does not impact the presentation or interpretations.

Minor points

>> Mean +/- s.d. should be amended to read Mean +/- 1 s.d. in all figure legends.

OK thank you. We have made the edits.

Figure 2B

>>The asterisk annotating the chymotrypsin digests are difficult to see - and will be too small when the figure is reduced for publication. The pink and green asterisks in particular do not seem to always lines up with a digested species. Some level of quantification (i.e. a histogram) for the appearance/disappearance of the C1 species for would assist the reader.

Thank you. We include a quantification of the C1 band in supplementary Figure 4 for a representative experiment.

Page 3

"Aptx mutants are sensitive to the 4NQO (Deshpande et al 2009)"

>> Remove the word 'the' from the above sentence.

Ok thank you. We have edited this sentence.

Figure 2D / Figure 2E

>> Both panels from the methyl-met NMR experiments should be plotted on the same x-axis limits (1.4 - 2.3 ppm) and both axes labelled consistently (i.e. include 16 and 20 on y-axis)

Ok thank you, good point. We have fixed this figure.

Figure 3C and 3D

>>Data for the wild-type protein should always be presented as the first bar in the histogram. It would also aid the reader to plot the histograms with a consistent order (i.e. not rank order) to allow a simple comparison. Perhaps using the clustering of mutation types presented later in the manuscript?

Ok thank you. We feel there are merits to ordering the mutants as the referee suggests, as well as how we presented these in the original submission. We have included the "consistent order" as suggested now as additional figure panels in the supplementary information (Supplementary Figure 6).

Figure 3D

>>There are no error bars associated with this histogram. Is this a simple oversight? - as the figure legend indicates that three technical replicates were preformed.

Yes, this was an oversight. We now include error bars and reporting in the legend.

Fig.2D/Fig.2E

Page 9

"Similar to core cavitation mutants of BRCA1(Williams et al, 2003; Williams et al 2004)"

>> Are all three self-citations fully justified / warranted here?

Thank you, this was a mistake. Williams et al 2001 is extraneous, and has been removed.

Supplementary Figure 4B

>> Typo in table header - "Differnence"

Thank you, edited.

2nd Editorial Decision

29 May 2018

Thank you for submitting your revised manuscript for our consideration. It has now been seen once more by two of the original reviewers (see comments below), and I am happy to inform you that they are satisfied with the revisions and now unconditionally recommend publication in The EMBO Journal.

Referee #1

The authors have adequately addressed my concerns.

Referee #3

With the amendments made to the manuscript, and with the supporting statements made in the rebuttal document, I am happy to recommend that the manuscript should be accepted for publication in The EMBO Journal.

Corresponding Author Name: Dr. R. Scott Williams

Manuscript Number: EMBOJ-2017-98875R